

# Ozonolysis of $\alpha$-phellandrene, Part 1: Gas- and particle-phase characterisation

Felix A. Mackenzie-Rae[1], Tengyu Liu[3,4,*], Wei Deng[2,3], Xinming Wang[2,3], Sandra M. Saunders[1], Zheng Fang[3,4], and Yanli Zhang[2,3]

[1]School of Chemistry and Biochemistry, The University of Western Australia, Crawley WA 6009, Australia
[2]Center for Excellence in Regional Atmospheric Environment, Institute of Urban Environment, Chinese Academy of Sciences, Xiamen 361021, China
[3]State Key Laboratory of Organic Geochemistry and Guangdong Key Laboratory of Environmental Protection and Resources Utilization, Guangzhou Institute of Geochemistry, Chinese Academy of Sciences, Guangzhou 510640, China
[4]University of Chinese Academy of Sciences, Beijing 100049, China
[*]now at: City University of Hong Kong

*Correspondence to:* X. Wang (wangxm@gig.ac.cn)

**Abstract.**

The ozonolysis of $\alpha$-phellandrene, a highly reactive conjugated monoterpene largely emitted by Eucalypt species, is characterised in detail for the first time using a smog chamber at the Guangzhou Institute of Geochemistry, Chinese Academy of Sciences. Gas-phase species were monitored by a proton-transfer-reaction time-of-flight mass spectrometer (PTR-TOF), with

yields from a large number of products obtained, including formaldehyde ($5-9\%$), acetaldehyde ($0.2-8\%$). glyoxal ($6-23\%$), methyl glyoxal ($2-9\%$), formic acid ($22-37\%$) and acetic acid ($9-22\%$). Higher *m/z* second-generation oxidation products were also observed, with products tentatively identified according to a constructed degradation mechanism. OH yields from $\alpha$-phellandrene and its first-generation products were found to be $35 \pm 12\ \%$ and $15 \pm 7\ \%$ respectively, indicative of prominent hydroperoxide channels. An average first-generation rate coefficient was determined as $1.0 \pm 0.7 \times 10^{-16}\ \mathrm{cm^3\ molecule^{-1}\ s^{-1}}$

at 298 K, showing ozonolysis as a dominant loss process for both $\alpha$-phellandrene and its first-generation products in the atmosphere. Endocyclic conjugation in $\alpha$-phellandrene was also found to be conducive to the formation of highly condensible products, with a large fraction of the carbon mass partitioning into the aerosol phase, monitored with a scanning mobility particle sizer (SMPS) and a high-resolution time-of-flight aerosol mass spectrometer (AMS). Nucleation was observed almost instantaneously upon ozonolysis, indicating the rapid formation of extremely low volatility compounds. Particle nucleation was

found to be suppressed by the addition of a Criegee scavenger, suggesting that stabilised Criegee intermediates are important for new particle formation in the system. Aerosol yields ranged from $25-174\%$ dependant on mass loadings, with both first- and second-generation products identified as large contributors to the aerosol mass. Effective density ranged from $1.29-1.90$ $\mathrm{g\,cm^{-3}}$. The aerosol oxidation state was also found to be dependent on mass loadings, with parametrisation of the bulk mass indicating a large contribution of highly functionalised low- and semi-volatile organic compounds to the aerosol phase. With a

high chemical reactivity and aerosol forming propensity $\alpha$-phellandrene is expected to have an immediate impact on the local environment to which it is emitted, with ozonolysis therefore likely to be an important contributor to the significant blue haze and frequent nocturnal nucleation events observed over Eucalypt forests.



# 1 Introduction

Biogenic sources dominate the global emission budget of volatile organic compounds into the atmosphere, with monoterpenes accounting for a significant fraction of nonmethane hydrocarbons emitted (Guenther et al., 1995; Schurgers et al., 2009; Guenther et al., 2012; Lathière et al., 2006; Sindelarova et al., 2014). Considering source strength, estimated to be 30 - 127

$\mathrm{Tg\,C\,year^{-1}}$, along with high chemical reactivity (Calvert et al., 2000; Atkinson and Arey, 2003), monoterpenes are thought to play an important role in the chemistry of the atmosphere; influencing its oxidative capacity, the tropospheric ozone budget and by producing secondary organic aerosol (SOA) with impacts to both health and climate (Hoffmann et al., 1997; Griffin et al., 1999a; Chung and Seinfeld, 2002; Hallquist et al., 2009; Pye et al., 2010). Indeed the ozonolysis of monoterpenes is thought to be one of the major sources of SOA in the atmosphere (Griffin et al., 1999b; Ortega et al., 2012).

Consequently the gas-phase reaction of ozone with monoterpenes has been the focus of numerous studies, both experimental, with focus on gas-phase kinetics and particle formation, properties and composition (e.g. Bateman et al., 2009; Berndt et al., 2003; Griffin et al., 1999a; Herrmann et al., 2010; Lee et al., 2006; Ma et al., 2007; Pathak et al., 2007; Saathoff et al., 2009; Shilling et al., 2008, 2009; Walser et al., 2008); and theoretical, utilising state-of-the-art computational methods (Zhang and Zhang, 2005; Nguyen et al., 2009). Collectively, research has gone a long way to understanding the mechanism and product dis-

tributions of monoterpene ozonolysis, and provided important insights into SOA precursors and production. Accurate chemical mechanisms for the reaction of specific monoterpenes with ozone have since been developed (Camredon et al., 2010; Jenkin, 2004; Leungsakul et al., 2005), whilst more general parameterisations for gas-phase reactions (Jenkin et al., 1997; Saunders et al., 2003) and SOA formation (Odum et al., 1996; Donahue et al., 2006; Stanier et al., 2008) have been implemented into chemical transport models.

Nevertheless research shows, when considered as a whole, significant variability in gas-phase oxidation products and SOA yields between different monoterpenes, highlighting the unique impact different monoterpenes can have on regional atmospheric chemistry. It is therefore important that individual monoterpene variability be accounted for in developing accurate gas- and particle-phase models. Nonetheless current literature has predominantly focussed on a small number of major monoterpenes (*e.g.* $\alpha$-pinene, $\beta$-pinene, limonene). One monoterpene for which respectively little is known is $\alpha$-phellandrene. One of

the most reactive monoterpenes, $\alpha$-phellandrene (Fig. 1) has been identified as a major constituent of extracts from various Eucalypt species (Jacobs and Pickard, 1981; Boland et al., 1991; Pavlova et al., 2015; Maghsoodlou et al., 2015), the world's most widely planted hardwood trees (Myburg et al., 2014). During day-to-day activities and processes these Eucalypt species, such as *Eucalyptus microtheca* and *Eucalyptus viminalis*, emit $\alpha$-phellandrene into the atmosphere, with $\alpha$-phellandrene likely contributing to the intense and frequent particle nucleation events observed over Eucalypt forests–a process already believed to

be caused by monoterpene oxidation (Suni et al., 2008; Lee et al., 2008; Ortega et al., 2009, 2012). In the indoor environment $\alpha$-phellandrene can be found as an additive to household cleaning products, detergents and air fresheners (e.g. Eucalypt themed products), with the European EPHECT project reporting $\alpha$-phellandrene at a concentration of $16.7\,\mathrm{\mu g\,m^3}$ in a study of a passive air freshener in a $1\,\mathrm{m^3}$ room after 5 hours (Stranger, 2013). Maisey et al. (2013) reported similar maximum concentrations of $\alpha$-phellandrene in Australian dwellings.





The rate constant of $\alpha$-phellandrene with ozone has been measured in a number of studies with results spanning an order of magnitude (Grimsrud et al., 1975; Atkinson et al., 1990; Shu and Atkinson, 1994), with a rate constant of $3.0 \times 10^{-15}$ ($\pm 35\%$) $\mathrm{cm^3\,molecule^{-1}\,s^{-1}}$ favoured (Calvert et al., 2000). High chemical reactivity likely makes ozonolysis a dominant loss process for $\alpha$-phellandrene in the atmosphere, however experimental information regarding reaction products is limited to OH radical

yields, measured by Herrmann et al. (2010) to be $26 - 31\%$ and $8 - 11\%$ for the ozonolysis of the two double bonds, and acetone yields, which were reported by Reissell et al. (1999) to be minor ($< 2\%$). Recently however the reaction mechanism was investigated theoretically for the first time by Mackenzie-Rae et al. (2016), who mapped the potential energy surface to first-generation products.

With a theoretical foundation, this study aims to experimentally characterise the reaction of $\alpha$-phellandrene with ozone

in detail for the first time by exploring and characterising both the gaseous- and particle-phases, with the impact of Criegee scavengers and $NO_2$ on the system also addressed. In doing so the impact of a highly reactive and potentially important monoterpene will be parametrised.

## 2 Materials and Method

### 2.1 Experimental set-up and procedure

Eleven dark $\alpha$-phellandrene ozonolysis experiments were conducted using the indoor smog chamber facility at the Guangzhou Institute of Geochemistry, Chinese Academy of Sciences (GIG-CAS). A complete description of the facility and chamber setup is given in Wang et al. (2014). Briefly, the GIG-CAS smog chamber consists of a $30\ \mathrm{m^3}$ fluorinated ethylene propylene (FEP) reactor housed inside a temperature controlled room. The reactor was flushed with purified dry air for at least 48 hours prior to each experiment, until no residual hydrocarbons, $O_3$, $NO_x$ or particles were detected. Two Teflon-coated fans located inside the

reactor ensure rapid homogenization of introduced species. Liquid reactants were vaporised via injection into a heating system similar to that of gas chromatography, before being carried by nitrogen gas through FEP Teflon lines into the reactor. Ozone was generated using a commercial ozone generator (VMUS-4, Azco Industries Ltd), with pure oxygen feed gas. Initial mixing ratios of the reactants varied between 10 and 175 ppb for $\alpha$-phellandrene (Aldrich Chemical Company, Inc., USA) and between 56 and 500 ppb for $O_3$, with all experiments conducted under dry conditions (Table 1). $\alpha$-phellandrene was injected prior to

admission of $O_3$ into the chamber, with $O_3$ added stepwise in experiments 7 and 10. Anhydrous cyclohexane (Sigma-Aldrich, 99.5%) was added in all but two experiments, in sufficient quantity to scavenge $> 95\%$ of OH radicals (Aschmann et al., 1996; Herrmann et al., 2010), whilst formic acid (J&K Scientific Ltd., 98%) was added to experiments 6 and 7 to ascertain the role of stabilised Criegee Intermediates (CIs) (Bonn et al., 2002; Winterhalter et al., 2009). Prior to $O_3$ addition in experiment 11, 385 ppb of $NO_2$ was added through a septum installed in one of the injection ports using a gas-tight syringe. All experiments had

2.5 μL of acetonitrile injected as a dilution tracer, with the top frame of the reactor periodically lowered to maintain a positive pressure differential inside the reactor. Experimental run times ranged from $205 - 305$ minutes.





## 2.2 Characterisation of gas- and particle-phases

Volatile organic compounds (VOCs) were measured online with a commercial proton-transfer-reaction time-of-flight mass spectrometer (PTR-TOF 2000, Ionicon Analytik GmbH, Austria) (Jordan et al., 2009; Graus et al., 2010), using $H_3O^+$ reagent ions. For data collected in the first 7 experiments in Table 1, the PTR-TOF drift tube was operated at 2.2 mbar and 60°C, with

a drift tube field of 600 V cm$^{-1}$ (E/N = 136 Td). Significant fragmentation was observed under this regime, with a drift tube voltage of 484 V cm$^{-1}$ (E/N = 112 Td) found to be optimal (Supplementary Information S.2). The refined operating conditions were then used for experiments 8 – 11. PTR-TOF spectra were collected at a time resolution of 2 seconds. Data were processed using the PTR-TOF Data Analyzer (Müller et al., 2013), with 30 spectra averaged to improve counts of trace species. A generic $H_3O^+$ rate constant of $2 \times 10^{-9}$ cm$^3$ s$^{-1}$ was used for conversion into ppb, except for those species where experimental or

theoretical data exists (Cappellin et al., 2012; Tani, 2013).

  Gas-phase $O_3$ and $NO_x$ were measured online using dedicated monitors (EC9810 and 9841T, Ecotech, Australia), which were calibrated regularly using a Thermo Scientific Model 146i multi-gas calibrator unit. In all experiments, excluding number 11 where it is added, $NO_x$ concentrations were negligible ($< 1$ ppb). The $O_3$ analyser experienced significant interference (had a false bias) from $\alpha$-phellandrene, which was corrected for using PTR-TOF measurements.

Particle number size distributions were measured online with a scanning mobility particle sizer (SMPS; TSI Incorporated, USA) (Wang and Flagan, 1990), consisting of an electrostatic classifier (TSI 3080) fitted with a TSI 3081 differential mobility analyser and condensation particle counter (TSI 3775). Sheath and aerosol flow rates were 3.0 and 0.3 L min$^{-1}$ respectively, with voltage inside the DMA varied exponentially from -10 V to -9950 V every 240 seconds to provide a mobility spectrum over particle diameters 14 – 750 nm. Higher moment size distributions were calculated by assuming spherical particles

(Wiedensohler et al., 2012).

  A high-resolution time-of-flight aerosol mass spectrometer (AMS; Aerodyne Research Incorporated, USA) was used to measure particle chemical composition in real-time (Jayne et al., 2000; DeCarlo et al., 2006). The AMS was operated in the high sensitivity V-mode and high resolution W-mode, switching between modes every 2 minutes. AMS data were analysed in Igor Pro 6.2 (Wavemetrics) using the ToF-AMS data analysis toolkits Peak Integration by Key Analysis (PIKA) and Sequential

Igor Data Retrieval (SQUIRREL). Updates were made to the fragmentation table following a similar method to Chen et al. (2011), with a detailed discussion provided in the Supplementary Information (S.5). Conductive silicon tubes were used as sampling lines for the SMPS and AMS to reduce electrostatic losses of particles, whilst all other instruments had FEP Teflon feed lines. Losses of VOCs and particles in the transfer lines are estimated to be less than 5% (Liu et al., 2015).

## 3 Results and Discussion

The starting conditions for each experiment are listed in Table 1. The high reactivity of $\alpha$-phellandrene towards ozone results in reaction half lives that are similar to the mixing time of the reactor. Consequently only a lower bound of the concentration of ozone is known.



### 3.1 Gas-phase Analysis

#### 3.1.1 Peak Identification and Yields

Significant fragmentation was observed in the PTR-TOF upon injection of starting materials into a clean reactor. $\alpha$-phellandrene was detected at *m/z* 137 at $32-34\%$ depending on drift tube conditions, consistent with fragmentation observed in the PTR-MS

studies of Misztal et al. (2012) and Tani (2013) (Supplementary Information S.1). Acetonitrile was found exclusively at *m/z* 42, and remained constant throughout all experiments indicating that dilution effects in the reactor are negligible (e.g. Fig. 2). Despite having a lower proton affinity than water, cyclohexane was detected at *m/z* 85, although overall sensitivity is greatly reduced. The detection of cyclohexane is likely the result of termolecular reactions in the PTR-TOF (Smith and Španěl, 2005), with observed cyclohexane fragments listed in Table 2. Meanwhile in a separate characterisation experiment formic acid was

found at *m/z* 47, with minor fragments at *m/z* 48, 49 and 65 ($< 2\%$).

Observed interferences are expected to impact detection of $\alpha$-phellandrene's degradation products, biasing signals to lower *m/z*. Aldehyde, ketone, alcohol, ester and acid bearing compounds are known to dehydrate following protonation to yield a $MH^+(-H_2O)$ daughter ion (Smith and Španěl, 2005; Blake et al., 2006). Furthermore multifunctional carbonyl compounds can eject a second water molecule from nascent $MH^+$ ions yielding a $MH^+(-H_2O)_2$ daughter ion, whilst complex acid

bearing molecules have been observed to fragment via the loss of formic acid to produce $MH^+(-HCOOH)$ ions, and esters through ejection of -OR groups to yield $MH^+(-ROH)$ (Španěl et al., 1997; Španěl and Smith, 1998). Uncertainty arising from fragmentation prevents quantitative analysis for the majority of species, with standards neither available nor prepared. Nevertheless, Table 2 lists peaks routinely detected by the PTR-TOF across the 11 experiments. Note that *m/z* includes the addition of $H^+$.

Figure 2 shows that $\alpha$-phellandrene is rapidly oxidised upon ozonolysis, forming a number of product ions at low concentrations. Ignoring conformational isomerism, the ozonolysis of $\alpha$-phellandrene can yield four unique CIs (Mackenzie-Rae et al., 2016). For simplicity only the degradation of one possible CI is shown in Fig. 7, with detailed schematics of the remaining CIs provided in the Supplementary Information (S.4). Elucidating the mechanism of $\alpha$-phellandrene one expects initially to form a large range of first-generation products, however none of the product ions detected were observed to decrease over the course of

the chamber experiments, suggesting that all ions are likely second-generation. For example from the stabilised CIs one might expect an unsaturated keto-aldehyde or dialdehyde product (Figs. S.4.1 and S.4.2), analogous to pinonaldehyde from $\alpha$-pinene and limonaldehyde from limonene, to be detected at *m/z* 169. Indeed this signal was observed, but it continued to increase in concentration after $\alpha$-phellandrene was consumed, suggesting that the observed *m/z* 169 is not simply a direct product ion of $\alpha$-phellandrene. Other major first-generation product ions expected include *m/z* 185, which corresponds to a range of isomeric

species formed through either excited or thermalised CI re-arrangement reactions, whereby three oxygen atoms are added for no loss of carbon or hydrogen (e.g. acids, esters, epoxides, secondary ozonides) and *m/z* 155, which can be formed through radical transfer and subsequent CHO loss in the hydroperoxide channel (Mackenzie-Rae et al., 2016). Both these ions were detected in the PTR-TOF but again had concentrations which increased throughout the experiments, suggesting that they instead correspond to higher generation products. A similar phenomenon, whereby a distinct lack of first-generation products were ob-





served by a PTR-MS, occurred in the ozonolysis of $\alpha$-terpinene (Lee et al., 2006), a structurally similar endocyclic-conjugated monoterpene. In the study of Lee et al. (2006) first-generation products were observed for other monoterpene species, including 3-carene, $\alpha$-pinene, $\beta$-pinene, terpinolene and myrcene. It is possible then that for highly reactive monoterpenes such as $\alpha$-terpinene and $\alpha$-phellandrene, concentrations of first-generation products do not accumulate sufficiently during experiments

for gas-phase detection. However, as discussed in later sections, a simple rate study shows that residence lifetimes are sufficient, whilst analysis of saturation concentrations suggests that the majority of predicted first-generation products likely reside in the gas-phase. It is therefore uncertain as to why first-generation products of $\alpha$-phellandrene were not detected in this work, but suggests some large loss or removal process for these functionalised, unsaturated species either in the sample lines and/or during transfer into and detection by the PTR-TOF.

The highest product signal concentrations were observed for low $m/z$ species ($\leqslant$ C$_3$). Whether this is an accurate representation of the system or a systematic bias from fragmentation is unknown, however anecdotally, increased counts of low mass species were observed as the energy of the drift tube was raised suggesting that the latter does have some effect. Major peaks were found at $m/z$ 31, 45, 47, 59, 61 and 73, corresponding to formaldehyde, acetaldehyde, formic acid, glyoxal, acetic acid and methyl glyoxal respectively. Although acetone also resides at $m/z$ 59, based on the low gas-chromatographic yields reported in

Reissell et al. (1999) the signal is apportioned to glyoxal. As $\alpha$-phellandrene contains two double bonds, yields in this work were calculated as the slope of the least square regression between the change in concentration of the oxidation product and change in wall loss corrected ozone, as shown in Fig. 3; with ozone wall loss rates frequently characterised following the method describe in Wang et al. (2014). The average yield from sequential ozonolysis is therefore calculated, however in practice calculations are dominated by data points measured after the consumption of $\alpha$-phellandrene, with the data corresponding

to the initial reaction of $\alpha$-phellandrene comparably limited and often largely excluded to reduce errors associated with having a finite reactor mixing time. As discussed later this problem is navigated for OH radicals by using a higher PTR-TOF time resolution and measuring yields against $\alpha$-phellandrene consumption, however mixing ratios of other oxidation products are too low in the initial stages of the experiment to produce reliable yield data in this regime.

   The low molecular weight acids, formic and acetic, were both found to be produced with high yields. The fragmentation

pattern of acetic acid was determined in a separate calibration experiment, with 88% residing at $m/z$ 61 and the remaining mass distributed over $m/z$ 43, 62 and 79, corresponding to dehydration to the acylium ion, the $^{13}$C isotope and protonation by a water cluster respectively. Correcting for fragmentation, yields of formic and acetic acid were found to range from 22 – 37% and 9 – 22% respectively across the conducted experiments (Table 3). Yields of formic acid are considerably higher than what has been reported for the ozonolysis of other terpenes, whilst acetic acid yields are consistent with species containing an endocyclic

bond (Lee et al., 2006). The addition of NO$_2$ was found to reduce yields of both formic and acetic acid to 10 $\pm$ 2 and 5 $\pm$ 1 respectively, with O$_3$ losses through reaction with NO$_x$ accounted for. The addition of NO$_2$ therefore acts as an inhibitor to acidic group formation, likely by scavenging acyl peroxy radicals to form peroxyacyl nitrates (PANs). Alternatively NO$_2$ can impact the chemistry of the system by reacting with stabilised secondary ozonides (SOZs), although no changes in acid product yields were observed in the experiments where stabilised CIs were scavenged, indicating that this channel is negligibly

important in forming low molecular weight acids.





In characterising the PTR-TOF transmission curve acetaldehyde and all other oxygenated VOCs in the gas-standard (Ionicon, Analytik GmbH, Austria) showed no evidence of fragmentation. Therefore, assuming no fragmentation for the remaining oxidation products, provides yields of formaldehyde, acetaldhyde, glyoxal and methyl glyoxal of 5 – 9%, 0.2 – 8%, 6 – 23% and 2 – 9%. Nevertheless fragmentation of methyl glyoxal through CO loss in the PTR-TOF has been reported (Müller et al., 2012), which would simultaneously reduce its own yield whilst increasing the yield of acetaldehyde. A similar phenomenon is also expected of glyoxal, nonetheless acetaldehyde yields remain low and consistent with findings reported for other terpene species. Formaldehyde yields are consistent with other terpene species containing multiple internal double bonds, e.g. $\alpha$-humulene and $\alpha$-terpinene (Lee et al., 2006). The addition of a CI scavenger was found to have little impact on product distribution or yields. Meanwhile in experiment 11 yields of formaldehyde, acetaldhyde, glyoxal and methyl glyoxal were 1.2 $\pm$ 0.3 %, 0.41 $\pm$ 0.09 %, 7.6 $\pm$ 2 % and 2.1 $\pm$ 0.5 % respectively. The systematic reduction in product yields upon introduction of $NO_2$ is consistent with a shift away from these carbonyl containing species towards more nitrate containing products.

Heavier second-generation products routinely detected across experiments are listed in Table 2, with yields for a number of these products given in Table 4. The absence of a yield indicates that the peak was not detected by the PTR-TOF, which typically occurred for minor peaks in experiments with lower initial $\alpha$-phellandrene concentrations. Again no fragmentation was assumed in determining yields, although some ions do differ by common fragment mass amounts, suggesting that fragmentation may be important. For example $m/z$ 185 and 167, $m/z$ 129 and 111 and $m/z$ 115 and 97 all differ by 18 amu, suggesting that the latter masses could be dehydrated fragments. Whilst strong correlation ($R^2 > 0.99$) between these pairs of peaks is observed it is not consistent across the entire dataset, suggesting that there exists multiple contributors to the aforementioned signals. Similar instances are also observed for peaks separated by 28 amu (e.g. $m/z$ 143 and 115) and 46 amu (e.g. $m/z$ 185 and 139).

Calculated yields for these larger products were in general < 5%. Again the addition of $NO_2$ to the system had the effect of reducing product yields, although overall distribution remained similar and no new peaks were observed, indicating that ozonolysis products remain dominant. Proposed structures for some of these larger second-generation products along with plausible formation mechanisms are shown in Fig. 7, although it is possible that more than one product contributes to an observed oxidation product mass. A large number of products also remain unidentified, with their $m/z$ unable to be transcribed to plausible, mechanistically derived, structures.

Figure 2 shows that product signals at $m/z$ 31, 59, 73 and 87 show a sharp increase upon commencement of the reaction, suggesting that these products are formed directly from $\alpha$-phellandrene ozonolysis. Nevertheless a large fraction of the product mass for these ions is generated after $\alpha$-phellandrene consumption, indicating that yields are largely driven by contributions from second-generation species. Slower initial production of the remaining ions suggests that their formation is linked to consumption of first-generation products. Interestingly the peaks corresponding to the heaviest ions, $m/z$ 167, 169 and 185, have relatively constant temporal profiles lacking an accelerated increase whenever ozone is introduced; a feature that is apparent among all other ions. Their unique time profiles implies that they are derived from a source secondary to ozonolysis, such as gas-phase accretion reactions (Supplementary Information S.3).





Formation of prescribed products after the second ozonolysis is in agreement with the proposed degradation mechanism (Fig. 7), which predicts a number of both small and large species to form upon fragmentation of the carbon backbone. A large fraction of the smaller products come from decomposition of the 3-carbon system ($C_1$-$C_2$-$C_7$, Fig. 1) bridging the conjugated double bonds in $\alpha$-phellandrene, which segments from the rest of the molecule after the second ozone addition. For example, plausible

mechanisms can be traced to methyl glyoxal formation irrespective of the order of addition of ozone to the two double bonds, with the only prerequisite being that the first addition of ozone adds one carbonyl group to the $C_1$-$C_2$-$C_7$ system (e.g. Fig. 6). Subsequent decomposition of the $C_1$-$C_2$-$C_7$ Criegee biradical fragment can yield products including formaldehyde, formic acid and acetic acid. Meanwhile functionalisation of the larger 7-carbon system bridging the conjugated bonds in $\alpha$-phellandrene can give rise to a large number of heavier second-generation products. *m/z* 129 is assigned to 2-propan-2-ylbutanedial, which

can be formed from a number of pathways (e.g. Fig. 7). *m/z* 115 is assigned to 2-propan-2-ylpropanedial, which is formed if a CI from either addition participates in the hydroperoxide channel resulting in CHO fragmentation. Conversely if instead of fragmentation, stabilisation occurs after a 1,5-hydrogen shift, then the product detected at *m/z* 143 shown in Fig. 7 may form. In all instances detected second-generation products can be formed from a wide variety of predicted first-generation products independent of the order of addition of ozone to the two double bonds (Supplementary Information S.4). So whilst the complete

product distribution will likely consist of a myriad of species (Aumont et al., 2005), in practice the relatively small number of second-generation products detected likely dominate the final ozonolysis product distribution.

### 3.1.2 Determination of OH Yields

OH yields from the reaction of $\alpha$-phellandrene and its first-generation degradation products with ozone are listed in Table 5. No OH scavenger was added in experiments 9 and 11. The OH radical scavenger, cyclohexane, reacts with OH to form

both cyclohexanone and cyclohexanol (Atkinson et al., 1992; Berndt et al., 2003), with cyclohexanone (*m/z* 99) used as the OH radical tracer in this study. In a characterisation experiment 98% and 85% of cyclohexanone (Sigma Aldrich, 99.8%) was found to reside at *m/z* 99 when the PTR-TOF drift tube was operated at 112 and 136 Td respectively, with the remaining mass distributed over a dehydrated and cluster peaks at *m/z* 81, 116 and 117. A minor ozonolysis product is also detected at *m/z* 99 (Section 3.1.1), however the two peaks are resolvable in the PTR-TOF.

A major uncertainty in determining OH yields is the yield of cyclohexanone formed from the reaction of cyclohexane with OH radicals. Atkinson et al. (1992) reported the combined yield of cyclohexanone and cyclohexanol to be 0.55 $\pm$ 0.09, with cyclohexanone/cyclohexanol ratios typically ranging from 0.8 – 1.4 depending on the terpene investigated. In contrast, Berndt et al. (2003) reported a cyclohexanone yield of 0.53 $\pm$ 0.06. In this study, the OH-yield is based on the average of these two findings, with a cyclohexanone yield from the reaction of OH and cyclohexane of 0.41 $\pm$ 0.14 used. Scavenging is assumed

to be 95% efficient based on the volume of cyclohexane introduced into the reactor, with the error in this assumption thought to be minimal with respect to the inherent uncertainty in cyclohexanone yields. Background interference from cyclohexane, of which a small portion is oxidised by $O_2^+$ to cyclohexanone in the drift tube, is corrected for (Winterhalter et al., 2009).

OH yields for the initial reaction of ozone with $\alpha$-phellandrene were calculated from the slope of OH produced against $\alpha$-phellandrene reacted. As both *m/z* 99 and 137 are major signals in the PTR-TOF, spectra were not averaged during analysis





resulting in a 2 second time resolution. A characteristic OH-production time profile is shown in Fig. 4, which can be separated into three regions. The initial part of the experiment is characterised by a linear section, where $\alpha$-phellandrene is the primary source of OH radicals. The gradient obtained from linear regression in this regime is equivalent to the OH yield from ozonolysis of the first double bond in $\alpha$-phellandrene (Fig. 5a). The $\alpha$-phellandrene dominated regime is short-lived with respect to total

OH production time in the reactor, suggesting that first-generation products are also highly reactive and large producers of OH radicals. As the reaction proceeds, faster reacting first-generation products begin to contribute to the OH budget, whilst $\alpha$-phellandrene becomes increasingly less influential. This results in a gradual curve, until at which point essentially all $\alpha$-phellandrene has been consumed and a vertical path is traced, indicating that first-generation species are now dominating OH radical production. By plotting OH formation against $O_3$ consumption in the product dominated regime, the collective

first-generation product OH radical yield is obtained (Fig. 5b). Naturally this method is only applicable to those experiments where the product dominated regime is attained, which is why no OH radical yields are reported for first-generation products in experiments 8 and 10.

    The average OH yield for the reaction of the first double bond in $\alpha$-phellandrene across the 10 experiments was found to be $35 \pm 12$ %, whilst the average OH yield from the ozonolysis of the second reacting double bond was $15 \pm 7$ %. Both these

determined values are slightly higher than the values calculated in Herrmann et al. (2010), although they agree well within uncertainty limits. Whilst experimental methodology is similar, Herrmann et al. (2010) conducted their analysis under the assumption that the two double bonds in $\alpha$-phellandrene react at significantly different rates, such that 95% of $\alpha$-phellandrene reacts before first-generation products start to be consumed. However as Fig. 4 shows, first-generation products contribute to the OH radical budget considerably earlier than this. As a result Herrmann et al. (2010) is likely to have under-predicted OH

yields of first-generation products, inadvertently apportioning their contribution to $\alpha$-phellandrene, whose OH yields would subsequently be over-predicted. The method employed in this study is thought to provide a more accurate distinction between OH radical production from $\alpha$-phellandrene and its first-generation products. From the determined yields it can be concluded that the hydroperoxide channel does play an important role in the decomposition of $\alpha$-phellandrene by ozone, supportive of findings from a recent theoretical study (Mackenzie-Rae et al., 2016).

### 3.1.3    Modelling rate constants and OH yields

The conjugated system in $\alpha$-phellandrene provides two reactive sites for ozone addition. Based on analogy with rate constants from simpler alkenes, such as cyclohexene (k = $8.1 \times 10^{-17}$ cm$^3$ molecule$^{-1}$ s$^{-1}$) and 1-methyl-1-cyclohexene (k = $1.66 \times 10^{-16}$ cm$^3$ molecule$^{-1}$ s$^{-1}$) (Calvert et al., 2000), inductive effects are expected to make the methyl substituted double bond the more reactive addition site. However recent theoretical results suggest the contrary (Mackenzie-Rae et al., 2016),

with steric effects raising the energy barrier for entry to the more substituted double bond, resulting in addition to the less substituted double bond in $\alpha$-phellandrene being favoured. This finding is consistent with experimental evidence for isoprene, where methacrolein, not methyl vinyl ketone, is the favoured first generation product (Paulson et al., 1992; Grosjean et al., 1993; Aschmann and Atkinson, 1994; Rickard et al., 1999). Nevertheless the average energy difference for addition to the two double bonds is minor, with both entry channels expected to be important.





Given the high chemical reactivity of both double bonds in $\alpha$-phellandrene, it is interesting to investigate the reactivity of first-generation products. The following reaction parametrisation was therefore constructed to determine the average rate constant of ozone with all first-generation species.

$$\alpha-\text{phellandrene} + O_3 \xrightarrow{k_1} FG + xOH$$

$$FG + O_3 \xrightarrow{k_2} SG + yOH$$

$$O_3 \xrightarrow{k_3} wO_3$$

where FG represents all first-generation products, SG all second-generation products, $wO_3$ is ozone lost to the reactor walls and x and y are stoichiometric coefficients representing OH yields from each reaction step. The rate constant for the reaction of ozone with a-phellandrene ($k_1$) was constrained to the literature value (Calvert et al., 2000), whilst a first-order ozone wall loss rate of $k_3 = 2 - 8 \times 10^{-6} \, \text{s}^{-1}$ was used based on a number of calibration experiments. Remaining parameters, namely x, y and $k_2$, were varied to optimise model performance. The reaction scheme was solved using the online numerical integrator AtChem (https://atchem.leeds.ac.uk/) for all experiments, barring 9 and 11 due to the unconstrained influence of $NO_2$ and/or OH radicals.

Figure 8 shows the results of the simulation of ozone consumption and OH production for three different experiments, with optimised parameters for each experiment given in Table 5. Considering simplicity the model performs surprisingly well. Based on all experiments the average simulated rate constant for the reaction of first-generation products with ozone was $k_2 = 1.0 \pm 0.7 \times 10^{-16} \, \text{cm}^3 \, \text{molecule}^{-1} \, \text{s}^{-1}$. Although the ozonolysis of first-generation products is around 30 times slower than that of $\alpha$-phellandrene, it is still faster than the ozonolysis of the monoterpenes $\alpha$-pinene, $\beta$-pinene, sabinene, 3-carene and $\beta$-phellandrene (Calvert et al., 2000). Using a typical background tropospheric ozone mixing ratio of 30 ppb, atmospheric lifetimes ($\tau$) of $\alpha$-phellandrene and its first-generation products can be estimated by:

$$\tau_i = \frac{1}{k_i[O_3]} \tag{1}$$

The atmospheric lifetime of $\alpha$-phellandrene is therefore $\tau_1 \sim 7.5$ minutes whilst the average lifetime of first-generation products is calculated to be $\tau_2 \sim 3.75$ hours. Both $\alpha$-phellandrene and its first-generation products therefore have a relatively short atmospheric lifetime with respect to ozone and are unlikely to be involved in long-range transport phenomena. Instead complete saturation likely occurs in the chemical environment to which $\alpha$-phellandrene is emitted, thus impacting the local radical, acid and SOA budgets. Interestingly increasing the ozone concentration to conditions found in chamber experiments results in first-generation product lifetimes of the order of tens of minutes, which is more than sufficient for gas-phase detection. The inability to detect first-generation products is therefore indicative of an underlying sampling or detection issue.

OH production is additionally included in the model to assist in parametrising yields from $\alpha$-phellandrene and the average of its first-generation products. Experimental assessment of OH yields carries an inherent uncertainty, in that linear regression was used to fit data belonging to a segment of a curve (Fig. 5), with information in the 'combination' section notionally discarded. The parametrisation employed therefore allows for a more complete description, although mechanistic simplicity renders the



results far from quantitative. Instead its purpose is to both validate experimental findings and allow further constraints to be placed on OH production from the $\alpha$-phellandrene system.

The average modelled OH yields are $54 \pm 10$ % and $13 \pm 5$ % for $\alpha$-phellandrene and its average first-generation products respectively. The model suggests $\alpha$-phellandrene makes a greater contribution to the OH budget than what was calculated experimentally, whilst yields from all first-generation products are consistent with experimental measurements. Overall net OH radical production is greater from the model than experimental results suggest. The source of this discrepancy is likely the limited data used in calculating $\alpha$-phellandrene's OH yields, with a large proportion of $\alpha$-phellandrene consumed in the 'combination' section. Given the $\alpha$-phellandrene dominated regime generally lasted around 2–3 minutes, which is comparable to the mixing time of the reactor, it is entirely possible that OH production in this regime was not characterised well. The experimental finding for OH radical production from $\alpha$-phellandrene of $35 \pm 12$ % is therefore recommended as a lower bound.

## 3.2 Particle-phase Chemistry

### 3.2.1 SOA Formation

First- and second-generation ozonolysis products are highly functionalised, polar species with high molecular weights. It is therefore expected that they should make a significant contribution to the aerosol phase through gas-particle partitioning (Pankow, 1994; Odum et al., 1996). To assess this, the saturation vapour concentration ($C^*$, $\mu g \, m^{-3}$) (Donahue et al., 2006, 2012) of each species was calculated. Vapour pressures were estimated using the Extended Aerosol Inorganic Model (E-AIM) (Clegg et al., 2008), using the structure based estimator of Nannoolal et al. (2004) for boiling points coupled with Moller et al. (2008) for vapour pressures. This method has been compared extensively with other estimation techniques (Barley and McFiggans, 2010; O'Meara et al., 2014). Activity coefficients were calculated using the UNIversal Functional Activity Coefficient (UNIFAC) method (Fredenslund et al., 1975). The saturation vapour concentrations calculated are shown in two-dimensional volatility oxidation space in Fig. 9.

The majority of predicted first-generation and detected second-generation gas-phase products are classified as intermediate volatility compounds (IVOCs) (Donahue et al., 2012). As IVOCs they are considered to have quite low vapour pressures, but nonetheless reside almost exclusively in the gas-phase. Of the proposed species, only the first-generation acids (e.g. Fig. 7) are classified as semi-volatile organic compounds (SVOCs), a classification given to those species which are expected to have sizeable mass fractions in both the gaseous and aerosol phase.

Nevertheless rapid aerosol formation is observed upon reaction of $\alpha$-phellandrene and ozone, as shown by sharp increases in particle number (dN/dlogDp) and volume (dV/dlogDp) concentrations (Fig. 10). With no aerosol seed, nucleation must be driven by supersaturation of condensible species. Donahue et al. (2013) argues that nucleation occurs through compounds that have extremely low volatility (ELVOC, $C^* < 3 \times 10^{-4} \, \mu g \, m^{-3}$). For the ozonolysis of other monoterpenes, ELVOC formation has been proposed to occur through gas-phase accretion reactions (Bateman et al., 2009; Heaton et al., 2009; Camredon et al., 2010) and autoxidation processes (Ehn et al., 2014; Jokinen et al., 2015). Meanwhile to condense onto fresh aerosol, but



not homogeneously nucleate, vapours need to have saturation concentrations in the $10^{-3} - 10^{-2}$ µg m$^{-3}$ range (Donahue et al., 2011; Pierce et al., 2011), placing them in the low volatility organic compound bin. Formation of these compounds can be explained through conventional gas-phase chemistry (Donahue et al., 2011). It is therefore evident that the simple mechanistic overview provided to explain formation of gas-phase products in Section 3.1.1 and in Mackenzie-Rae et al. (2016)

is insufficient to account for aerosol observations, with more complex reactions required to develop species of sufficiently low vapour pressure for both particle nucleation and growth.

  The maximum number of particles inside the reactor occurs within the first few minutes of the reaction commencing (time resolution of the SMPS), with a small average particle diameter (∼40 nm). Rapid nucleation is consistent with the findings of Jokinen et al. (2015) who, based on limonene and $\alpha$-pinene, concluded that endocyclic biogenic VOCs are efficient ELVOC

producers upon ozonolysis. Coagulation of the newly-formed aerosol decreases the number of particles, whilst further partitioning of low volatility oxidation products increases the volume, with maximum aerosol concentration attained around 30 minutes into each experiment. After this point irreversible wall losses supersedes gains from partitioning, with the volume, and hence mass of aerosol decreasing inside the reactor.

  In the early stages of experiments, the number concentration is a useful proxy for measuring the amount of nucleation

occurring in the system. As Fig. 11 shows, the addition of a Criegee scavenger systematically reduces initial particle number concentrations, concurrent with a shift of SOA to larger diameters. These changes suggest a reduction in the number of SOA nucleating agents (Bonn et al., 2002), implying that the reaction of $\alpha$-phellandrene stabilised CIs, whether uni- or bi-molecular, are important in forming ELVOC and IVOC compounds, whilst ruling out the reaction of stabilised CIs with formic acid as a nucleating mechanism. This finding is consistent with experimental literature that is now building around stabilised CIs as

a source of new particle formation; whether through intramolecular SOZ formation (Bonn et al., 2002), bimolecular reaction with first-generation products (Bateman et al., 2009), or oligomer formation through reaction with peroxy radicals (Sadezky et al., 2006, 2008) or hydroperoxides (Sakamoto et al., 2013). Processes such as these would all be precluded by the addition of formic acid to the system.

  Assuming spherical particles, effective aerosol densities were calculated by comparing distributions of vacuum aerodynamic

and electric mobility diameters, using the AMS and SMPS respectively (DeCarlo et al., 2004; Katrib et al., 2005; Kostenidou et al., 2007). Results are listed in Table 6. The average density across all experiments was found to be $1.49 \pm 0.2$ g cm$^{-3}$, indicating that the aerosol exists in a solid or waxy state. This value is consistent with the SOA density found in the ozonolysis of other monoterpenes under similar conditions, which typically range from $1.15 - 1.73$ g cm$^{-3}$ (Bahreini et al., 2005; Kostenidou et al., 2007; Saathoff et al., 2009; Shilling et al., 2009). Because the particles are potentially non-spherical, the

quoted effective density represents a lower bound of the true $\alpha$-phellendrene SOA density, with error from assuming spherical particles expected to be less than 10% (DeCarlo et al., 2004; Bahreini et al., 2005). It is noted that the densest aerosol was produced in experiment 11, which had NO$_2$ added, although one experiment is insufficient for reliable conclusions. The aerosol density was found to be insensitive to a range of other experimental parameters, including starting $\alpha$-phellandrene and ozone concentrations, aerosol loading, aerosol oxidation state and the presence of CI scavengers. These findings are in contrast to





studies conducted on $\alpha$-pinene (Shilling et al., 2009) and $\beta$-caryophyllene (Chen et al., 2012), in which particle density was found to decrease as chamber aerosol loadings increased in accordance with changes in aerosol oxidation state.

Aerosol densities were used to convert SMPS volume concentrations into mass loadings ($\mu g\,m^{-3}$). Wall loss effects were corrected for by assuming a size-independent first-order loss process (Pathak et al., 2007), by modelling data at the end of each

experiment after gas-aerosol partitioning had reached equilibrium. Calculated wall loss rate constants, which ranged from 0.32 $-0.79\,h^{-1}$, were then applied to correct mass loading data for respective experiments. This way, differences between individual chamber simulations are accounted for. Determined wall loss rates are consistent with those found for $\alpha$-pinene ozonolysis in the chamber (Wang et al., 2014).

The same method was used to correct V-mode AMS data, with results given in the Supplementary Information (S.6). Clus-

tering of points around the 1:1 line in Fig. S.6.1 indicates general agreement between mass loadings calculated using the AMS and SMPS (Shilling et al., 2008). Nevertheless density corrected SMPS data is preferred in this work, primarily because the AMS is known to suffer from transmission losses caused by particles bouncing off the vapouriser, and to a lesser extent, shape dependent collection losses whilst focussing the particle beam (Matthew et al., 2008; Slowik et al., 2004; Huffman et al., 2005). Whilst it is noted that the SMPS does not measure particles with diameters larger than 750 nm, as shown in Fig. 10, this

shortcoming is expected to have minimal impact on reported yields in this work (Wiedensohler et al., 2012).

Wall loss corrected mass loadings for each experiment are given in Table 6, along with fractional aerosol yields (Y). The fractional aerosol yield is defined as the amount of organic particulate matter that is produced ($\Delta M_o$, $\mu g\,m^{-3}$) for a given amount of precursor VOC reacted ($\Delta HC$, $\mu g\,m^{-3}$) (Odum et al., 1996), and provides a convenient way of assessing the bulk aerosol-forming potential of an individual VOC. Utilising the gas/particle partitioning framework, aerosol yield can be

described as a function of organic aerosol mass concentration (Pankow, 1994; Odum et al., 1996):

$$Y = \frac{\Delta M_o}{\Delta HC} = \Delta M_o \sum_i \frac{\alpha_i K_{om,i}}{1 + K_{om,i}\Delta M_o} \tag{2}$$

where $\alpha_i$ is the stoichiometric factor and $K_{om,i}$ the temperature-dependent equilibrium partitioning constant of product i. Whilst a large number of products are expected to contribute to the particle phase, in general a two-product model provides a satisfactory fit, with higher orders found to be superfluous (Odum et al., 1996; Griffin et al., 1999a; Hoffmann et al., 1997).

A characteristic yield plot is given in Fig. 12. SOA yield is best fit using the parameters, $\alpha_1 = \alpha_2 = 0.60\pm0.09$ and $K_{om,1} = K_{om,2} = 0.022\pm0.01\,m^3\,\mu g^{-1}$. The fitted constants offer little physical insight, other than perhaps the average of all $\alpha$ and $K_{om}$ values, but nonetheless can be used in regional and global modelling (Chung and Seinfeld, 2002).

Figure 12 shows $\alpha$-phellandrene produces a large amount of aerosol upon ozonolysis compared to other monoterpenes (Wang et al., 2014; Saathoff et al., 2009; von Hessberg et al., 2009). Formation of the necessary semi-volatile organic compounds is

likely driven by the presence of two highly reactive endocyclic double bonds, with functionalisation rather than fragmentation dominating for the first addition (Lee et al., 2006). Both experiments, where a CI scavenger was added, lie below the fitted yield curve, strengthening the argument for stabilised CIs as a source of condensible products. Cyclohexane has been shown to reduce SOA yields in ozonolysis experiments (Bonn et al., 2002; Keywood et al., 2004; Saathoff et al., 2009), although no such effects were observed in this study.





The addition of ozone to first-generation products in general fragments the molecule, but in doing so increases relative oxygen content. Thus the relative contribution of first- and second-generation products to SOA is empirically difficult to predict. Figure 13 shows SOA mass as a function of $\alpha$-phellandrene reacted, producing time-dependent aerosol growth curves. In all experiments where $\alpha$-phellandrene was completely consumed, dominant vertical growth profiles are traced. This increase

in aerosol mass after complete consumption of parent hydrocarbon is characteristic of compounds with more than one double bond (Ng et al., 2006), and suggests that when formed, second-generation products make an important contribution to the total aerosol mass. It is therefore likely that a large number of second-generation species fall in the IVOC or SVOC category in Fig. 9.

Whilst concentrations of precursors are somewhat elevated in experiments compared to ambient conditions, results nonethe-

less show $\alpha$-phellandrene ozonolysis products to be heavily involved in both particle nucleation and growth processes. In polluted environments (e.g. inner city forests, consumer products) a high SOA yield results in a large fraction of $\alpha$-phellandrene partitioning into the particle phase irrespective of gas-phase loadings. For example at the current World Health Organisation PM$_{2.5}$ air quality guideline (25 $\mu g \, m^{-3}$, World Health Organization (2006)), aerosol yield from $\alpha$-phellandrene ozonolysis would be $\sim$43%. Meanwhile a strong nucleation potential makes $\alpha$-phellandrene ozonolysis a strong candidate to help explain

the intense and frequent nocturnal nucleation events observed in Eucalypt forests (Lee et al., 2008; Suni et al., 2008), which is already believed to be caused by monoterpene oxidation products (Ortega et al., 2009, 2012). Currently aerosol formation from gas-phase precursors in clean environments is a globally important, yet poorly-quantified phenomenon. Detailed modelling studies are required to establish the relative importance of $\alpha$-phellandrene in these scenarios, although preliminary evidence suggests it is likely a contributor to nucleation events in regions where it is emitted.

**3.2.2   SOA Composition**

Resolution in the W-mode of the AMS is sufficient to unambiguously identify chemical formulae of detected ions (DeCarlo et al., 2006; Aiken et al., 2007), with a typical mass spectra shown in Fig. 14. For clarity only major peaks are depicted (relative intensity > 0.001). Ions are formed in the AMS using electron impact ionisation (70 eV). The high-energy collisions result in a large amount of fragmentation, although charge location and fragmentation patterns can generally be predicted from the

functional groups present (McLafferty and Turecek, 1993). Examples are shown for a detected first- and second-generation gas-phase products containing acid, aldehyde and ketone functionality, with proposed fragment ions labelled in the mass spectra. However the complexity of aerosol produced, along with an unknown number of fragmentation pathways, including the possibility of charge migration and other internal rearrangements, makes it exceedingly difficult to obtain clear structural information about SOA constituents. For this reason it is useful to use the AMS data to analyse the bulk properties of aerosol,

to gain further insight into the system.

Two ions commonly used as markers for oxidation of OA are $CO_2^+$ (*m/z* 44) and $C_2H_3O^+$ (*m/z* 43) (Takegawa et al., 2007; Kroll et al., 2009; Chhabra et al., 2010). In general, low-volatility oxygenated organic aerosol (LV-OOA) produces mass spectra dominated by the mass fragment $CO_2^+$, whilst semi-volatile oxygenated organic aerosol (SV-OOA) produce spectra with $C_2H_3O^+$ as the dominant ion. These ions are thought to be produced by organic acids and carbonyls respectively, with





the ratio of $C_2H_3O^+$ to $CO_2^+$ therefore providing a useful metric for assessing the degree of functionalisation in OA. During the chamber ozonolysis of $\alpha$-phellandrene, it was found that the relative importance of $CO_2^+$ is greatest at the start of the experiment, with its fractional contribution with respect to $C_2H_3O^+$ decreasing as aerosol mass loadings increase (Fig. 15). A small shift back to $CO_2^+$ is then observed in the latter stages of experiments.

The observed trend is unsurprising, as without any aerosol seed the nucleating species would likely be highly functionalised, featuring a large amount of acidic groups. As the experiment proceeds and aerosol mass increases, more volatile species start to contribute to the growing OA mass (Pankow, 1994; Odum et al., 1996), which increases the relative amount of $C_2H_3O^+$. Once growth has ceased and SOA loadings decrease (e.g. wall loss, evaporation), a small shift back to the less volatile species, and so $CO_2^+$, is observed. The ratio of *m/z* 43/44 was found to be dependent on total aerosol mass inside the reactor, with

the highest ratios found in those experiments with the greatest aerosol loadings. This trend is visible in the three experiments shown in Fig. 15, supporting the notion of increased contribution of less volatile species at higher aerosol mass loadings.

    Bulk elemental composition can be estimated by averaging ion contributions across the entire mass spectrum (Aiken et al., 2007). Raw measured atomic ratios are converted to estimated ratios using the calibration factors of Aiken et al. (2008), namely $0.91 \pm 10\%$ for hydrogen-to-carbon (H/C), $0.75 \pm 31\%$ for oxygen-to-carbon (O/C) and $0.96 \pm 22\%$ for nitrogen-to-carbon

ratios respectively, thus accounting for chemical biases in fragmentation.

    Temporal trends in elemental ratios (e.g. Fig. 16) are consistent with the narrative constructed by the *m/z* 43/44 ratio. Initial aerosol formation and growth is driven by highly oxygenated species however, as the OA medium grows and less functionalised species begin to partition, the overall oxidation state rapidly decreases, as seen by a drop in O/C and respective rise in H/C ratios. Once gas-particle partitioning slows and aerosol-loss processes dominate, there is a shift in equilibrium with the more

volatile aerosol constituents evaporating back into the gaseous-phase. It can therefore be concluded that many of the SOA products generated during the chamber ozonolysis of $\alpha$-phellandrene in this study are semi-volatile (Donahue et al., 2012). Nitrogen containing species were found to make little contribution to the aerosol formed in experiment 11, with an average N/C $\approx 0.002$ during the experiment.

    The average oxidation state of carbon ($\overline{OS}_c$) in aerosol comprising of carbon, hydrogen and oxygen was parameterised by

Kroll et al. (2011) as:

$$\overline{OS}_c \approx 2O/C - H/C \tag{3}$$

The definition ignores the effects of peroxides, whose oxygen atoms carry an oxidation state of -1, nevertheless serves as a useful metric for representing the degree of oxidation of organic species in complex aerosol mixtures. Figure 17 shows that $\overline{OS}_c$ decreases from -0.61 to -1.00 as the particle loading increases from 21.5 to 658.1 $\mu g\,m^{-3}$, again suggesting a strong link

between mass loading and degree of functionalisation (Shilling et al., 2009). The fastest change in $\overline{OS}_c$ is observed to occur at lower mass loadings. Calculated $\overline{OS}_c$ classifies the aerosol formed throughout the campaign as SV-OOA (Kroll et al., 2011), consistent with numerous monoterpene + $O_3$ chamber experiments (Bateman et al., 2009; Aiken et al., 2008; Shilling et al., 2009; Chhabra et al., 2010; Chen et al., 2011).





The average elemental composition across all experiments was found to be $CH_{1.57}O_{0.36}$. The high oxygen content with little respective loss of hydrogen is consistent with the ozonolysis mechanism, and indicates a large contribution from organic acids, alcohols and/or peroxides. As products only contain carbon, hydrogen and oxygen with triple bonds unlikely, the double bond index (DBE) can be used to further parameterise the OA (Bateman et al., 2009):

$$DBE = 1 - H/2 + C \qquad (4)$$

The DBE is equal to the total number of C=C bonds, C=O bonds and rings in the molecule. Assuming 10 carbons, the DBE value for the average SOA composition is 3.2. This value is consistent with hypothesised first-generation products (Figs. S.4.1 and S.4.2), which typically retain a double bond and have at least two oxygen-containing functional groups. For the larger second-generation species, which are thought to have 6 – 7 carbons, the DBE from the average elemental composition is 2.3

– 2.5. Assuming products are non-cyclic and saturated, the DBE indicates that second-generation species contributing to the SOA have on average 2 – 3 C=O bonds. Of course the impact of heterogeneous reactions, oligomerisation and many more complex processes thought to occur inside SOA cannot be discounted (Hallquist et al., 2009).

A Van Krevelen plot of the entire dataset is given in Fig. 18. The impact of CI scavengers, cyclohexane and $NO_2$ on OA in Van Krevelen space is observed to be minor. The important parameter, again, was found to be aerosol mass loadings. The

impact of SOA loadings on bulk aerosol metrics must now be stressed, with caution advised when translating chamber results that operate under relatively high aerosol loadings to an ambient setting, for which lower mass loadings are much more typical. With respect to Van Krevelen space, changes in aerosol mass loading results in vertical shifts, consistent with a change in oxidation state. Ozonolysis reactions are unique, as oxygen can be added, and condensible products formed, with no loss (and possibly gain) of hydrogen. Because of this, generic functionalisation lines used to characterise reactions in Van Krevelen space

(Heald et al., 2010; Chhabra et al., 2011) are not applicable.

It is evident from Fig. 18 that the majority of gas-phase species detected using the PTR-TOF (Table 2) have a lower O/C ratio compared to what is measured for the bulk of the aerosol, further confirming their primary residence in the gas-phase. It is therefore unlikely that any of the detected gas-phase species are substantially contributing to the generated aerosol, which instead is dominated by more functionalised products. Whilst it is likely that species comprising the OA are also present in the

gas-phase, they just exist below or are lost in detection by the PTR-TOF. Indeed the presence of a filter prior to the PTR-TOF inlet may hinder detection of less volatile species, as elevated levels of OA on the filter may coax species into partitioning.

The carbon mass balance for each experiment (Fig. 19) was calculated by summing the gas-phase yields of all product ions, assuming a carbon number of 6 for unidentified products, with SOA yields, whose carbon content was determined from elemental ratios measured in each experiment. The carbon balance ranged from 25 – 131%. General losses in the system affect

the ability to close the carbon mass balance for most experiments, with performance worse in those experiments with lower starting $\alpha$-phellandrene concentrations due to an inability to detect minor gas-phase products (Table 4). It is evident from Fig. 19 that, despite having lower yields, heavier gas-phase products make a larger contribution to the carbon mass balance than lighter species such as formaldehyde, glyoxal, formic acid and acetic acid, whose nominal yields are higher. Meanwhile experiment 4 had a carbon mass balance exceeding 100%, which is thought to be the result of an erroneously high SOA yield



(Fig. 12). It is immediately obvious from the carbon mass balance that in all experiments, a large fraction of $\alpha$-phellandrene partitions into the aerosol phase upon ozonolysis, exemplifying the impact $\alpha$-phellandrene can have on SOA growth upon introduction into an environment. Currently the species comprising SOA generated from $\alpha$-phellandrene ozonolysis remain unidentified, however a complete analysis of filter samples collected during these experiments is underway, in preparation for a follow-on publication.

## 4 Conclusions

The reaction of $\alpha$-phellandrene with ozone was studied in depth for the first time through 11 chamber experiments. In the gas-phase, only signals with increasing temporal profiles were detected by the PTR-TOF indicative of second-generation products. Of these, small species ($\leq C_3$) were found to be produced in the highest yields, namely formaldehyde ($5 - 9\%$), acetaldehyde ($0.2 - 8\%$). glyoxal ($6 - 23\%$), methyl glyoxal ($2 - 9\%$), formic acid ($22 - 37\%$) and acetic acid ($9 - 22\%$), with yields of all products suppressed by the addition of $NO_2$. Despite having lower yields, heavier second-generation products were found to make a larger contribution to the carbon mass balance. A small number of second-generation products were tentatively identified based on a constructed gas-phase mechanism, with some evidence for gas-phase dimers existing. Experimental OH-radical yields of $35 \pm 12\%$ and $15 \pm 7\%$ for $\alpha$-phellandrene and its first-generation products are in good agreement with those reported in Herrmann et al. (2010) and show the hydroperoxide channel to be an important pathway, with model output from a simple reaction parametrisation suggesting the yield from $\alpha$-phellandrene to be a lower bound. Meanwhile modelling provides a rate coefficient of $1.0 \pm 0.7 \times 10^{-16} \, \mathrm{cm^3 \, molecule^{-1} \, s^{-1}}$ for the average reaction of first-generation products with ozone at 298 K. This equates to an atmospheric lifetime of around 3.75 hours, higher than many other monoterpenes, and suggests that complete saturation of $\alpha$-phellandrene likely occurs in the environment to which it is emitted.

$\alpha$-phellandrene was found to form a large amount of aerosol upon reacting with ozone. A homogeneous nucleation burst of fresh aerosol was observed in all experiments within the first few minutes of the reaction, indicating a rapid formation of ELVOC species. Addition of a CI scavenger inhibited nucleation, suggesting that stabilised CIs are important precursors in forming compounds of low volatility in the system. The mechanism behind this remains unknown, although numerous pathways have been proposed in the literature for CIs from other alkenes. The average effective SOA density was determined to be $1.49 \pm 0.2 \, \mathrm{g \, cm^{-3}}$ with an oxidation state varying from 0.56 to 1.02 depending on mass loadings, with the highly functionalised aerosol therefore existing in a solid or waxy state. SOA growth curves show both first- and second-generation species contribute to the particulate phase, driving aerosol growth through to completion of the reaction. Using a two-product model, the SOA yield is best fit by parameters, $\alpha_1 = \alpha_2 = 0.60 \pm 0.09$ and $K_{om,1} = K_{om,2} = 0.022 \pm 0.01 \, \mathrm{m^3 \, \mu g^{-1}}$, with the SOA forming potential from $\alpha$-phellandrene ozonolysis greater than other monoterpenes previously investigated in the literature.

High radical, acid and SOA yields coupled with a high reactivity results in $\alpha$-phellandrene having an immediate and significant impact on its local environment. Indeed it appears likely that ozonolysis of $\alpha$-phellandrene contributes to the significant blue haze, and intense and frequent nocturnal nucleation events observed over Eucalypt forests. Characterisation and parametri-



sation of both the gaseous- and particle-phases formed from the ozonolysis of $\alpha$-phellandrene therefore betters our understanding on the impact of biogenic emissions, and begins to enable the inclusion of this potentially important monoterpene in future atmospheric models.

## 5   Data availability

5   A website dedicated to the smog chamber is currently under construction, which will include all chamber data. For the meantime original data pertaining to this work can be obtained upon request from Xinming Wang (wangxm@gig.ac.cn).

*Competing interests.*   The authors declare that they have no conflict of interest.

*Acknowledgements.*   Experiments were made possible through funding by the Strategic Priority Research Program of the Chinese Academy of Sciences (Grant No. XDB05010200); Ministry of Science and Technology of China (Grant No. 2016YFC0202204); and National Natural

10   Science Foundation of China (Grant No. 41530641/41571130031). The authors would also like to extend their thanks to Antoinette Boreave (IRCELYON) for assistance with operating the AMS.




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





**Figure 1.** $\alpha$-phellandrene – for clarity hydrogens are not shown, carbon atom labels referred to in main text.





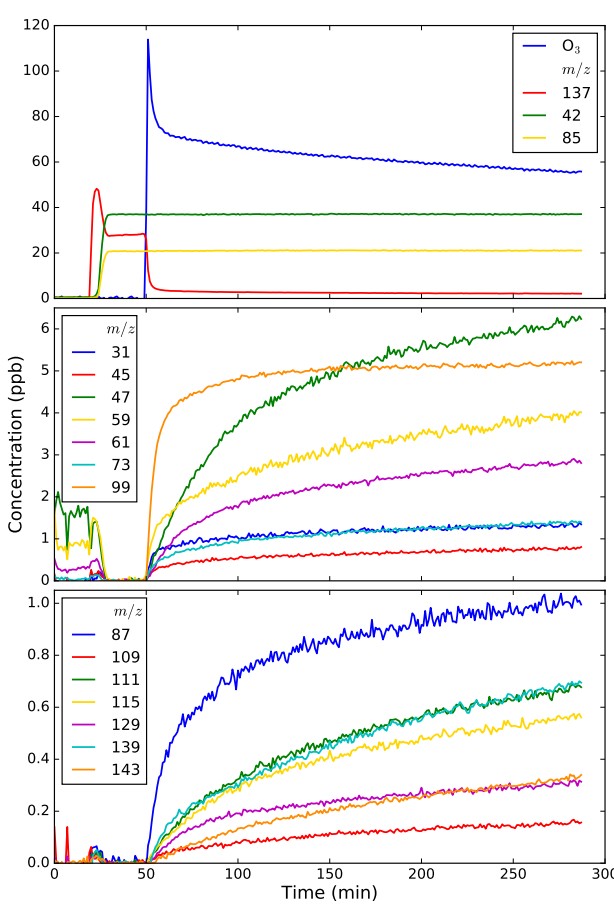

**Figure 2.** Time profiles of major species detected using the PTR-TOF during the ozonolysis of $\alpha$-phellandrene in experiment 5.





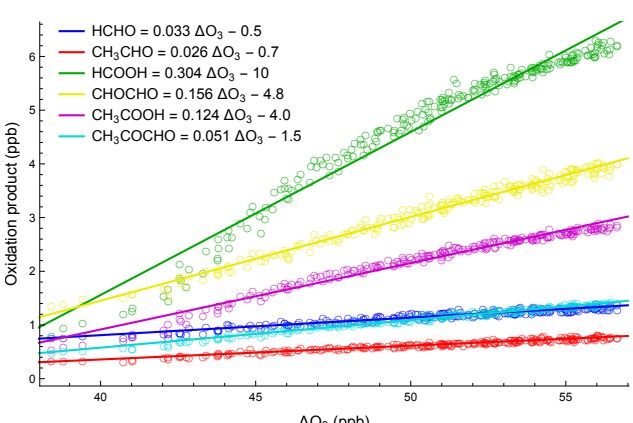

**Figure 3.** Determination of gas-phase product yields in experiment 5.





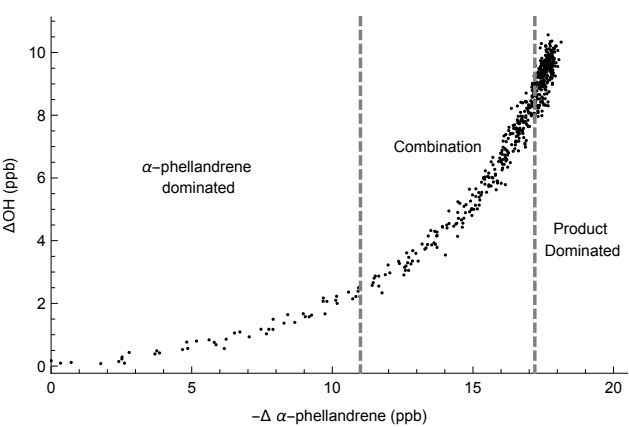

**Figure 4.** OH radical production versus $\alpha$-phellandrene consumption for the first 18 minutes of experiment 3.





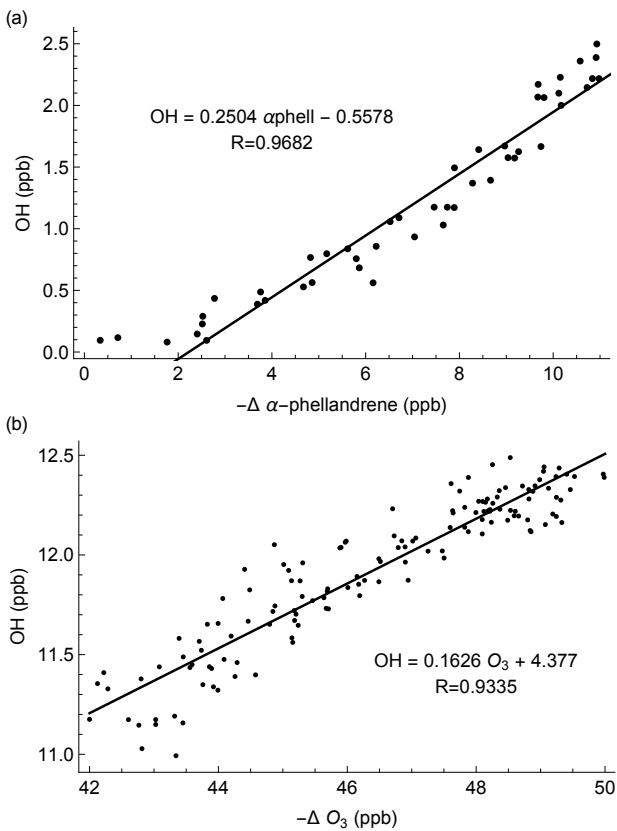

**Figure 5.** OH production from the (a) first and (b) second addition of ozone to $\alpha$-phellandrene in experiment 3 against $\alpha$-phellandrene and ozone consumption respectively.



**Figure 6.** Mechanism of $O_3$ addition to the proposed *m/z* 169 structures, yielding pairs of Criegee intermediates and carbonyl containing products.



**Figure 7.** Partial mechanism for the ozonolysis of α-phellandrene yielding product masses detected by the PTR-TOF. Other CIs and reaction pathways exist but are not shown here.



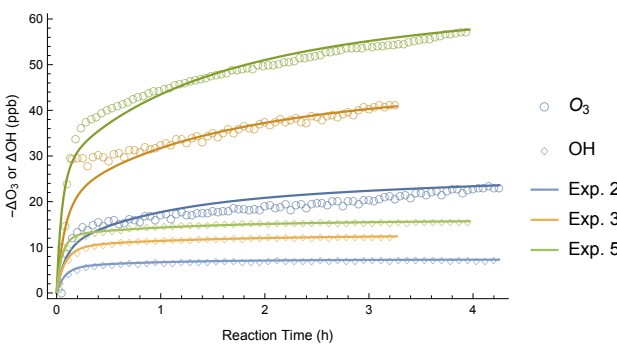

**Figure 8.** Plot of consumption of ozone and OH production against reaction time for experiments 2 (blue), 3 (yellow) and 5 (green). Experimental data are represented by open circles for O$_3$ and open diamonds for OH, whilst solid lines are modelled results using parameters in Table 5.





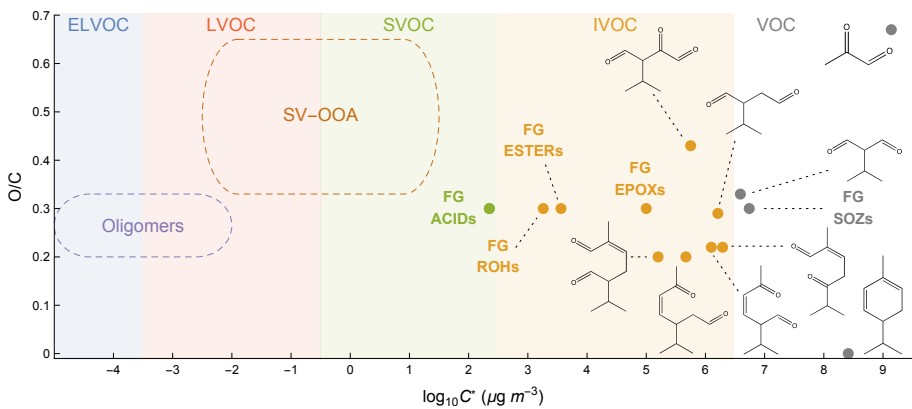

**Figure 9.** Predicted first-generation and detected second-generation products from the ozonolysis of $\alpha$-phellandrene in Donahue et al. (2006) space.





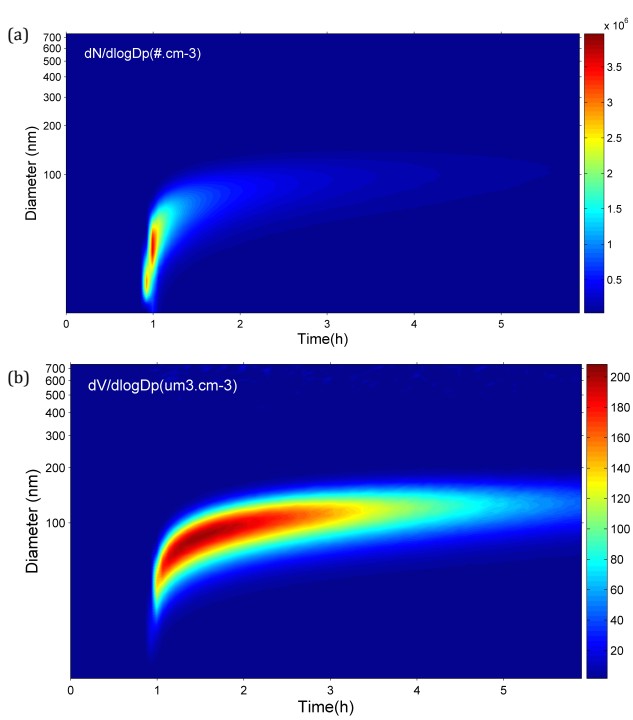

**Figure 10.** (a) Particle number $(\mathrm{cm}^{-3})$ and (b) volume $(\mathrm{\mu g\,cm}^{-3})$ size distributions for experiment 3.



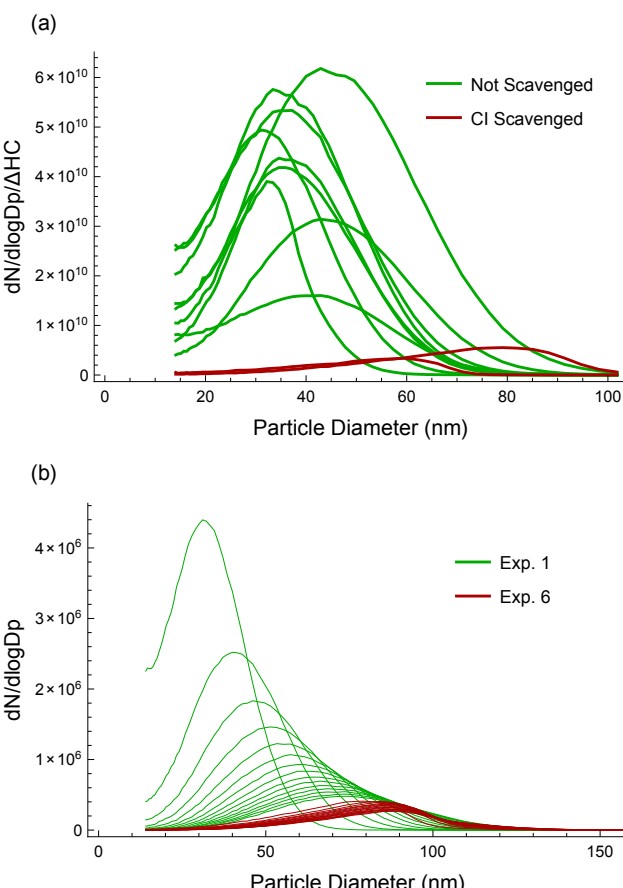

**Figure 11.** Impact of CI scavengers on particle nucleation shown by (a) peak particle number distributions scaled for the amount of $\alpha$-phellandrene reacted in all experiments and (b) particle number distributions evolution over the first hour of experiments 1 and 6.



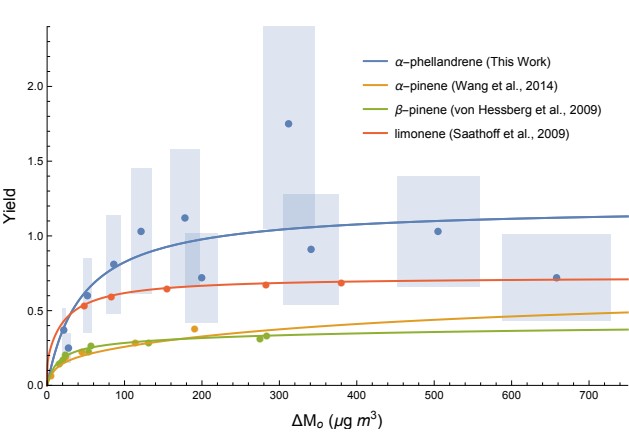

**Figure 12.** Comparison of SOA yield data for $\alpha$-phellandrene with other monoterpene ozonolysis experiments. Lines are the best two-product model fits (see text).





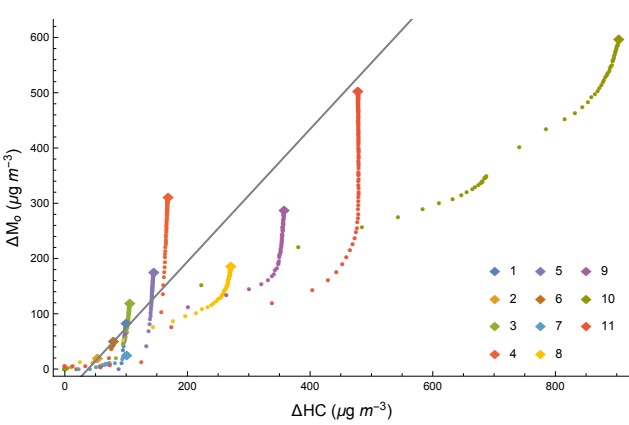

**Figure 13.** Time dependent SOA growth curves. Grey line is fitted yield curve and legend refers to experiment number.





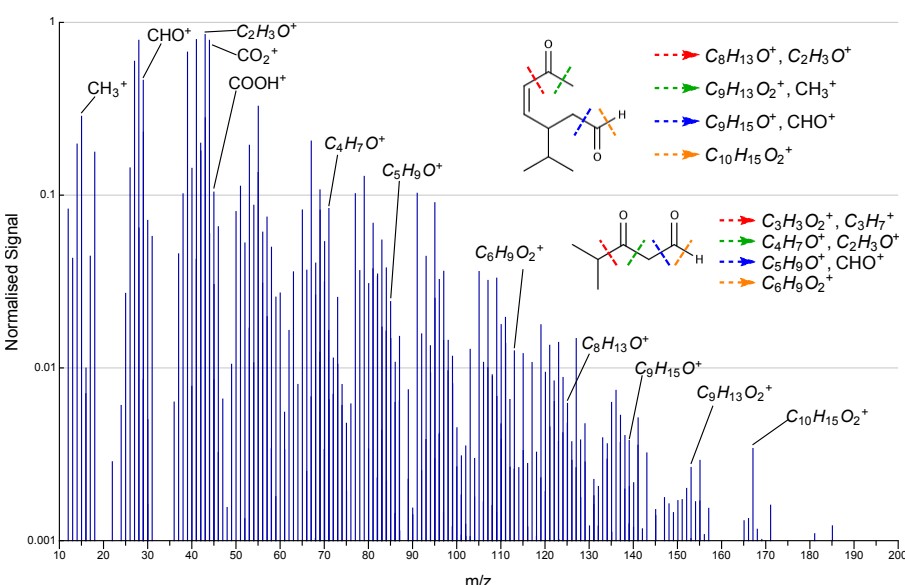

**Figure 14.** High-resolution AMS mass spectra at peak SOA loading from experiment 1. Inset are fragmentation routes of identified gas-phase products yielding detected ions.



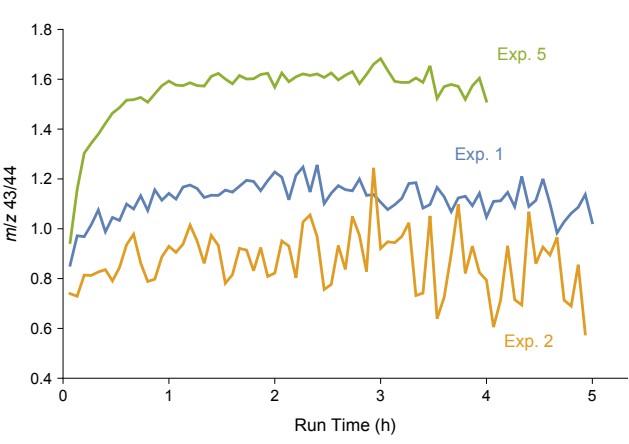

**Figure 15.** Ratio of *m/z* 43 ($C_2H_3O^+$) to *m/z* 44 ($CO_2^+$) signals in AMS spectra for three $\alpha$-phellandrene experiments.





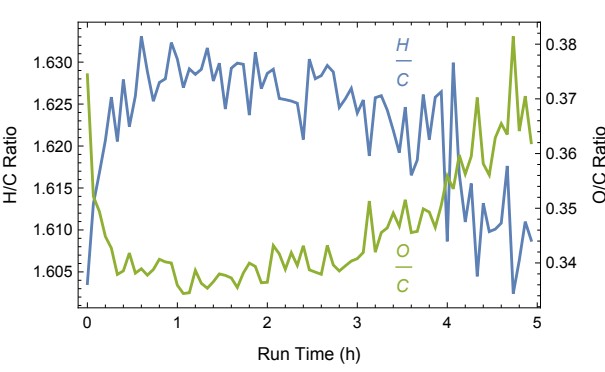

**Figure 16.** H/C and O/C ratios as a function of time for a typical $\alpha$-phellandrene ozonolysis experiment, number 4.



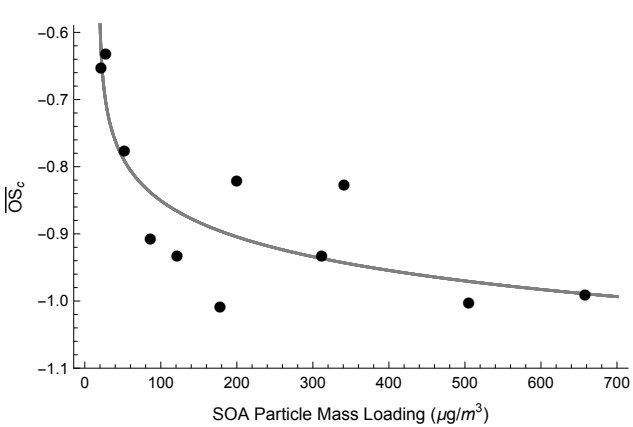

**Figure 17.** Average oxidation state of carbon for increasing SOA loadings generated through $\alpha$-phellandrene ozonolysis experiments, with general trend shown.





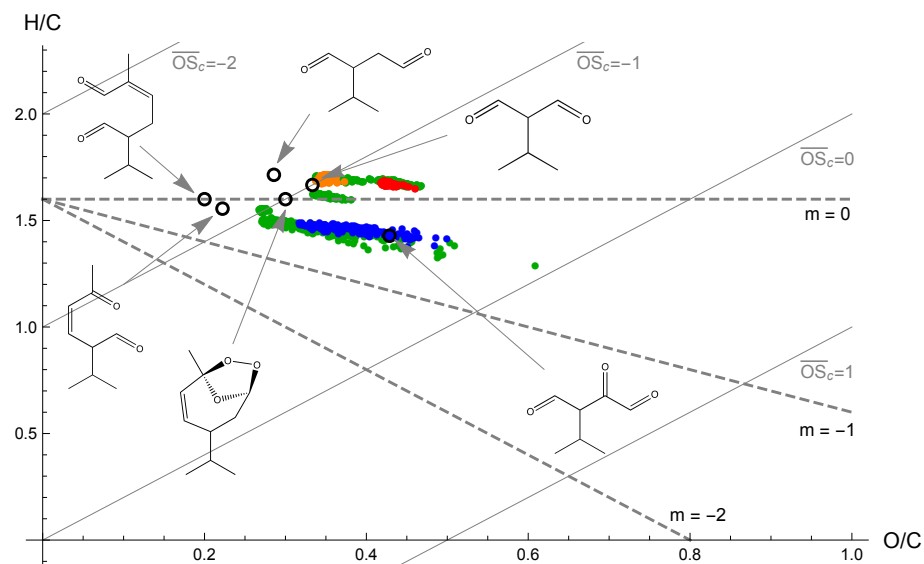

**Figure 18.** Van Krevelen plot. Blue dots are for experiments with a CI scavenger (6, 7), red dots for the experiment without cyclohexane (9), yellow dots for experiment with $NO_2$ added (11) and green dots for remaining experiments. Both predicted and detected gas-phase species are shown with open black circles.





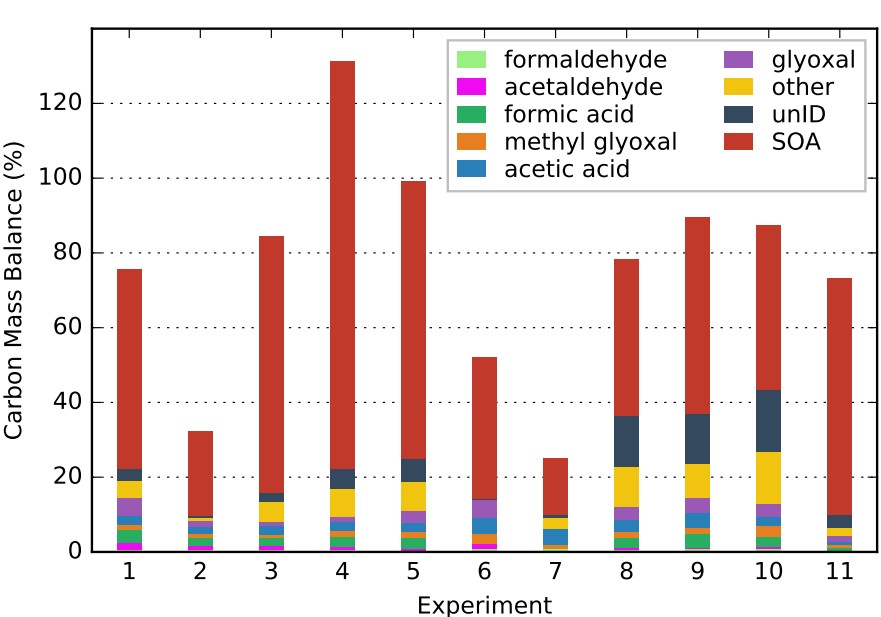

**Figure 19.** Carbon mass balance for each experiment.



**Table 1.** Starting conditions for $\alpha$-phellandrene chamber ozonolysis experiments.

| No. | Temperature (K) | Relative Humidity (%) | $\alpha$-phellandrene (ppb) | $O_3$ (ppb) | Additives[a] |
|-----|-----------------|-----------------------|------------------------------|-------------|--------------|
| 1 | $297.1 \pm 0.4$ | $2.5 \pm 0.6$ | $19 \pm 7$ | $> 259$ | Cyclohexane |
| 2 | $297.5 \pm 0.5$ | $2.1 \pm 0.7$ | $10 \pm 4$ | $> 86$ | Cyclohexane |
| 3 | $297.2 \pm 0.2$ | $2.3 \pm 0.6$ | $21 \pm 8$ | $> 83$ | Cyclohexane |
| 4 | $297.4 \pm 0.5$ | $2.2 \pm 0.9$ | $32 \pm 13$ | $> 193$ | Cyclohexane |
| 5 | $297.6 \pm 0.7$ | $1.8 \pm 0.4$ | $29 \pm 11$ | $> 114$ | Cyclohexane |
| 6 | $298.0 \pm 0.3$ | $1.6 \pm 0.1$ | $16 \pm 6$ | $> 470$ | Cyclohexane Formic Acid |
| 7 | $298.0 \pm 0.1$ | $1.9 \pm 0.2$ | $19 \pm 8$ | $> 499$ | Cyclohexane Formic Acid |
| 8 | $298.7 \pm 0.6$ | $5.2 \pm 0.2$ | $61 \pm 24$ | $> 56$ | Cyclohexane |
| 9 | $298.5 \pm 0.4$ | $4.9 \pm 0.4$ | $67 \pm 27$ | $> 101$ | – |
| 10 | $298.2 \pm 0.5$ | $4.8 \pm 0.3$ | $175 \pm 69$ | $> 174$ | Cyclohexane |
| 11 | $298.1 \pm 0.4$ | $4.5 \pm 0.2$ | $88 \pm 35$ | $> 132$ | $NO_2$ (385 ppb) |

[a] All experiments had acetonitrile (2.5 µL) added as a dilution tracer.



**Table 2.** Identified ions detected by the PTR-TOF. Refer to Figures 7 for product structures.

| | *m/z* | Formula | Assignment |
|---|---|---|---|
| Primary Signals | 21 | $H_3O^{18+}$ | Hydronium ion |
| | 37 | $(H_2O)_2H^+$ | Water cluster |
| | 55 | $(H_2O)_3H^+$ | Water cluster |
| Acetonitrile | 42 | $CH_3CNH^+$ | Acetonitrile |
| Cyclohexane | 28, 39, 40, 41, 42, 43, 44, 54, 55, 56, 57, 58, 67, 68, 69, 70, 82, 83, 84, 85, 86 | $C_6H_{12}H^+$ | Cyclohexane and fragments |
| | 81, 99, 100, 116, 117 | $C_6H_{10}OH^+$ | Cyclohexanone |
| | 83, 101 | $C_6H_{12}OH^+$ | Cyclohexanol[a] |
| Formic Acid | 47, 48, 49, 65 | $CH_2O_2H^+$ | Formic Acid |
| Acetic Acid | 43, 61, 62, 79 | $C_2H_4O_2H^+$ | Acetic Acid |
| α-phellandrene | 43, 67, 69, 79, 81, 82, 83, 91, 92, 93, 94, 95, 109, 119, 121, 135, 136, 137, 138, 139, 153 | $C_{10}H_{16}H^+$ | See Supplementary Information (S.1) |
| Ozonolysis Products | 31 | $CH_2OH^+$ | Formaldehyde |
| | 45 | $C_2H_4OH^+$ | Acetaldehyde |
| | 47 | $CH_2O_2H^+$ | Formic Acid |
| | 59 | $C_2H_2O_2H^+$ | Glyoxal |
| | 61 | $C_2H_4O_2H^+$ | Acetic Acid |
| | 73 | $C_3H_4O_2H^+$ | Methyl Glyoxal |
| | 87 | $C_3H_2O_3H^+$ | |
| | 115, 97 | $C_6H_{10}O_2H^+$ | Identified oxidation |
| | 129, 111 | $C_7H_{12}O_2H^+$ | products[b] |
| | 143 | $C_7H_{10}O_3H^+$ | |
| | 85[c], 99, 109, 125, 139, 155 | – | Unidentified oxidation products |
| | 167, 169, 185 | – | Gas-phase dimers |

[a] Winterhalter et al. (2009)

[b] Refer to Fig. 7

[c] Detected in experiments 9 and 11.



**Table 3.** Gas-phase molar yields (%) for major $\alpha$-phellandrene ozonolysis products.

| No. | formaldehyde | acetaldehyde | formic acid | glyoxal | acetic acid | methyl glyoxal |
|-----|--------------|--------------|-------------|---------|-------------|----------------|
| 1 | $6.9 \pm 2$ | $8.3 \pm 2$ | $37 \pm 9.0$ | $23 \pm 5$ | $13 \pm 3$ | $3.7 \pm 0.9$ |
| 2 | $5.9 \pm 1$ | $4.4 \pm 1$ | $24 \pm 6$ | $9.0 \pm 2$ | $9.0 \pm 2$ | $3.1 \pm 0.7$ |
| 3 | $5.9 \pm 1$ | $5.4 \pm 1$ | $22 \pm 5$ | $6.2 \pm 1$ | $12 \pm 3$ | $2.0 \pm 0.5$ |
| 4 | $5.0 \pm 1$ | $3.8 \pm 0.9$ | $28 \pm 6$ | $7.6 \pm 2$ | $11 \pm 2$ | $5.7 \pm 1$ |
| 5 | $3.3 \pm 0.8$ | $2.6 \pm 0.6$ | $30 \pm 7$ | $16 \pm 4$ | $12 \pm 3$ | $5.1 \pm 1$ |
| 6 | $7.0 \pm 2$ | $7.6 \pm 2$ | | $24 \pm 6$ | $22 \pm 5$ | $8.5 \pm 2$ |
| 7 | $8.7 \pm 2$ | $0.2 \pm 0.04$ | | | $22 \pm 5$ | $3.4 \pm 0.8$ |
| 8 | $5.4 \pm 1$ | $2.3 \pm 0.5$ | $28 \pm 6$ | $17 \pm 4$ | $16 \pm 4$ | $5.2 \pm 1$ |
| 9 | $7.5 \pm 2$ | $2.5 \pm 0.6$ | $35 \pm 8$ | $21 \pm 5$ | $20 \pm 5$ | $5.3 \pm 1$ |
| 10 | $7.9 \pm 2$ | $2.2 \pm 0.5$ | $29 \pm 7$ | $17 \pm 4$ | $13 \pm 3$ | $9.2 \pm 2$ |
| 11 | $1.2 \pm 0.3$ | $0.41 \pm 0.09$ | $10 \pm 2$ | $7.6 \pm 2$ | $5.0 \pm 1$ | $2.1 \pm 0.5$ |



**Table 4.** Minor gas-phase molar yields (%) for $\alpha$-phellandrene ozonolysis.

| No. | m/z 87 | m/z 97 | m/z 109 | m/z 111 | m/z 115 | m/z 129 | m/z 139 | m/z 143 |
|---|---|---|---|---|---|---|---|---|
| 1 | $2.5 \pm 0.6$ | $1.1 \pm 0.3$ | $0.58 \pm 0.1$ | $2.9 \pm 0.7$ | $3.3 \pm 0.8$ | | $3.4 \pm 0.8$ | |
| 2 | $2.5 \pm 0.6$ | $0.87 \pm 0.2$ | $0.19 \pm 0.04$ | | | | | |
| 3 | $3.2 \pm 0.8$ | $1.3 \pm 0.3$ | $0.61 \pm 0.1$ | $3.6 \pm 0.9$ | $3.0 \pm 0.7$ | | $2.3 \pm 0.5$ | |
| 4 | $2.5 \pm 0.6$ | $0.21 \pm 0.05$ | $0.74 \pm 0.2$ | $3.6 \pm 0.9$ | $3.0 \pm 0.7$ | $1.3 \pm 0.3$ | $4.4 \pm 1$ | $1.9 \pm 0.5$ |
| 5 | $3.5 \pm 0.8$ | $1.1 \pm 0.3$ | $0.78 \pm 0.2$ | $3.7 \pm 0.9$ | $3.0 \pm 0.7$ | $1.4 \pm 0.3$ | $3.7 \pm 0.9$ | $2.0 \pm 0.5$ |
| 6 | $0.13 \pm 0.03$ | | $0.45 \pm 0.1$ | | | | | |
| 7 | $0.75 \pm 0.2$ | | | | $2.4 \pm 0.6$ | | $1.7 \pm 0.4$ | $1.5 \pm 0.4$ |
| 8 | $5.4 \pm 1$ | $2.0 \pm 0.5$ | $1.1 \pm 0.3$ | $4.8 \pm 1$ | $3.1 \pm 0.7$ | $1.4 \pm 0.3$ | $4.2 \pm 1$ | $4.4 \pm 1$ |
| 9 | $2.3 \pm 0.5$ | $0.76 \pm 0.2$ | $0.68 \pm 0.2$ | $2.2 \pm 0.5$ | $3.0 \pm 0.7$ | $1.9 \pm 0.5$ | $3.8 \pm 0.9$ | $5.1 \pm 1$ |
| 10 | $7.4 \pm 2$ | $2.7 \pm 0.6$ | $0.69 \pm 0.2$ | $4.0 \pm 0.9$ | $2.7 \pm 0.6$ | $6.0 \pm 1$ | $3.9 \pm 0.9$ | $4.4 \pm 1$ |
| 11 | $0.12 \pm 0.03$ | $0.13 \pm 0.03$ | $0.15 \pm 0.03$ | $0.09 \pm 0.02$ | $0.91 \pm 0.2$ | $0.29 \pm 0.07$ | $0.40 \pm 0.09$ | $1.7 \pm 0.4$ |



**Table 5.** Measured and modelled OH radical yields and modelled rate constants for $\alpha$-phellandrene ozonolysis experiments.

| | $\alpha$-phellandrene | | | First-generation products | | |
|---|---|---|---|---|---|---|
| | $k_1$ ($10^{-15}$ cm$^3$ molecule$^{-1}$ s$^{-1}$) | Experimental OH Yield (%) | Modelled OH Yield (%) | $k_2$ ($10^{-16}$ cm$^3$ molecule$^{-1}$ s$^{-1}$) | Experimental OH Yield (%) | Modelled OH Yield (%) |
| 1 | 2.0 | $29 \pm 8$ | 48 | 0.7 | $10 \pm 2$ | 10 |
| 2 | 2.0 | $25 \pm 8$ | 55 | 2.0 | $27 \pm 5$ | 13 |
| 3 | 2.0 | $25 \pm 8$ | 48 | 1.5 | $16 \pm 4$ | 11 |
| 4 | 2.0 | $21 \pm 7$ | 46 | 0.6 | $10 \pm 2$ | 8 |
| 5 | 2.0 | $28 \pm 8$ | 45 | 1.0 | $11 \pm 3$ | 10 |
| 6 | 3.0 | $54 \pm 14$ | 67 | 0.3 | $20 \pm 4$ | 20 |
| 7 | 3.0 | $48 \pm 13$ | 65 | 1.0 | $10 \pm 3$ | 23 |
| 8 | 2.0 | $43 \pm 14$ | 68 | 2.0 | – | 10 |
| 10 | 2.0 | $47 \pm 11$ | 46 | 0.3 | – | 15 |
| Lit. | $3.0 \pm 1$[a] | $26 - 31$[b] | | | $8 - 11$[b] | |

[a] Calvert et al. (2000)

[b] Herrmann et al. (2010)





**Table 6.** Aerosol loadings, effective densities, oxidation states and yields for $\alpha$-phellandrene ozonolysis experiments.

| No. | Total SOA Mass[a] ($\mu g\,m^3$) | Density ($g\,cm^{-3}$) | $\overline{OS}_c$ | Yield (Y) |
|---|---|---|---|---|
| 1 | $86.1 \pm 9$ | $1.29 \pm 0.05$ | $-0.91 \pm 0.3$ | $0.81 \pm 0.3$ |
| 2 | $21.5 \pm 2$ | $1.32 \pm 0.06$ | $-0.65 \pm 0.2$ | $0.37 \pm 0.2$ |
| 3 | $121.3 \pm 13$ | $1.37 \pm 0.05$ | $-0.93 \pm 0.3$ | $1.03 \pm 0.4$ |
| 4 | $311.9 \pm 33$ | $1.57 \pm 0.05$ | $-0.93 \pm 0.3$ | $1.74 \pm 0.7$ |
| 5 | $178.0 \pm 19$ | $1.36 \pm 0.05$ | $-1.0 \pm 0.3$ | $1.11 \pm 0.5$ |
| 6 | $52.0 \pm 6$ | $1.38 \pm 0.05$ | $-0.77 \pm 0.3$ | $0.60 \pm 0.2$ |
| 7 | $27.6 \pm 3$ | $1.34 \pm 0.05$ | $-0.63 \pm 0.2$ | $0.25 \pm 0.1$ |
| 8 | $199.7 \pm 21$ | $1.69 \pm 0.06$ | $-0.82 \pm 0.3$ | $0.72 \pm 0.3$ |
| 9 | $341.0 \pm 36$ | $1.61 \pm 0.05$ | $-0.83 \pm 0.3$ | $0.90 \pm 0.37$ |
| 10 | $658.1 \pm 70$ | $1.60 \pm 0.05$ | $-0.99 \pm 0.3$ | $0.71 \pm 0.3$ |
| 11 | $504.9 \pm 53$ | $1.90 \pm 0.06$ | $-1.0 \pm 0.3$ | $1.02 \pm 0.4$ |

[a] Wall-loss corrected (Pathak et al., 2007).