# Peer review of "Ozonolysis of $\alpha$ -phellandrene, Part 1: Gas- and particle-phase characterisation"

_Atmospheric Chemistry and Physics, 2017_

## Referee Comment (RC1) · Anonymous Referee #1 · 28 Feb 2017

**General comments**

This paper presents experimental results on the formation of gaseous organic species, OH radicals and secondary organic aerosols (SOA) during a-phellandrene ozonolysis. Various dark ozonolysis experiments were performed in an indoor smog chamber facility (with or without OH scavengers, CI scavengers and NOx). Gas-phase species were monitored using a PTR-TOF-MS, SOA size distributions with a SMPS and aerosol compositions with an AMS. Measurements were used to calculate yields for a few gaseous organic species, OH yields from a-phellandrene and its first-generation products, and rate coefficient of first-generation products. Aerosol yields and effective density of SOA are also provided. The reactivity of a-phellandrene with ozone, as well as its impact on HOx and SOA formation has currently not been studied in the scientific community. This paper provides experimental data that are valuable to better understand the

environmental impact of biogenic compound ozonolysis and could therefore be of interest to the scientific community. However, the manuscript suffers from a lack of clarity, precision and discussion in several places. In particular, the current knowledge on a-phellandrene ozonolysis is not enough presented, the purpose of the selected type of experiments are not explained, the methodology used to calculate yields or kinetic constants should be clarified and conclusions reached by the author should be better justified by comparison with previous work. All of this make that even if the results may well be fully valid, an explanation of the methodologies and a justification of conclusions are needed to support the results. Major revisions are therefore required before publication in ACP.

**Specific comments**

The chemical reactions expected to occur during a-phellandrene ozonolysis according to the literature are never described rigorously in the manuscript. The discussion and the Figure in S4 should be presented before section 2 (and not in section 3 where it is difficult to make the difference between our knowledge, the coherence of the observations shown in this study and the novelty of the results). This figure should be used in section 3 to justify the detection of some species and could be coupled with in Fig. 7 to explain the formation of the "new" species detected in this study and with Fig. 6. The structure of a-phellandrene can also be presented from this figure, allowing to remove Figure 1 which is not really useful.

The various experiments are listed but the objective of each different type as well as the expected impact on the chemical system should be discussed (why several O3 injections, NO2 addition? what is the expected impact of an OH and a Criegee intermediate (CI) scavenger on the chemistry? how much of the CI is expected to be scavenged by the used amount of formic acid?). The observed influences of OH and CI scavengers, NO2 or the several O3 additions are not enough shown or discussed, questioning the interest of including exp. 6, 7, 9 and 11 in the paper. Do the authors see expected or unexpected differences in gaseous secondary organic and OH yields,

SOA formation. . .?

What is the typical volume of the chamber at the end of an experiment? (p.3 l.31)

Numerous figures and tables are discussed before or without being presented. Each Figure or Table should be clearly introduced in the text before being discussed.

The temporal evolution of the PTR-TOF-MS signals presented in Fig 2 should be described in the text at the beginning of the section 3.1.1. In particular, Why are all concentrations at around 35 min going to 0 except m/z = 42 and 137? Why does the precursor's concentration decrease at the same time? The m/z 42 signal is not impacted. Why? Is the linear decrease of O3 (after 70 min) due to wall loss?

In OH scavenged experiment, the reactivity of the system is expected to stop when the two a-phellandrene double bonds have each been reacting with O3. PTR-TOF-MS measurements show an increase of all the m/z signals (also for small molecules), except for the m/z assigned to the precursor and the inert gas. Some of these m/z could be assigned to first generation products (and also maybe second generation?) expected to be formed during scavenged a-phellandrene + O3 experiments. These first and also second generation species are expected to stop growing when no double bond remain in the molecule considering the known chemistry. The authors claim at several places in the paper that the observed increase suggests that these compounds are second-generation or higher generation species (e.g. p.5 l.25, l.35 and below in the paper) or that this unique profile implies that they derived from a source secondary to ozonolysis, such as gas-phase accretion (e.g. p7 l.35., table 2, S3...). The authors should be careful in drawing their conclusions. Why an increasing signal makes the species a second-generation compound? If this increasing signal comes from the formation of dimers, why is an increase also observed for the low m/z signals (m/z = 47, 59, 61...)? The hypothesis of some chamber wall artefacts is never discussed in the paper. Have the author already tested the impact of chamber teflon walls on the off-gassing of radical and/or organic species in the gas phase? Could it be a possible

explanation of the increasing m/z signals?

Experimental results are used to provide various yields. The yields are however not always clearly defined in the paper. For secondary organic species (SOC), Y would be usually defined as: $Y = \Delta[SOC]/\Delta[COV]$. Here Y is calculated from the amount of O3 that has reacted as $Y = \Delta[SOC]/\Delta[O3]$. Has the $\Delta[O3]$ to be divided by a factor of 2 to correspond to the $\Delta[COV]$ or not? Has this method been used previously? How do you deal with experiments without OH scavenger or with various additions of O3? The initial concentration of O3 is not known. Does this have an impact on the calculated yields? Looking at Fig. 3, no data is used at the beginning of the experiment when the system is the most reactive, i.e. before $\Delta[O3]=40$ ppb. Why? The intercept of the regression lines is different of 0. Why? The all length of the experiment is used to optimize regression lines and therefore to get the yield. 4/5 of these measured data correspond to a decrease of O3 concentration due to wall loss and an increase of secondary organic species which is difficult to understand. Is that not an issue to keep these points for the optimization?

Two OH yields are calculated, one for a-phellandrene and another one for its first-generation products using the methodology presented in Herrmann et al. (2010). The methodology used here should be presented using the Herrmann et al. (2010) reference with the two reactions p.8 l.34, before the presentation of the OH yield results. How do you define the end of the a-phellandrene dominated zone (Fig. 4)? Why is the intercept of the regression line different of 0 in Fig 5.a.? The intercept could be 0 with a good correlation coefficient if the end of the a-phellandrene dominated zone stops before. How do you define the end of the product dominated zone (Fig. 4)? Is that not when the observations are difficult to understand and when we do not look at ozonolysis? In Fig 5.b, shouldn't the OH and O3 concentration variations be looked starting from the beginning of the product dominated zone (and not from the beginning of the experiment) and why is the intercept of the regression line far from 0 ?

The kinetic constant for the reaction of the secondary products with O3 was optimized

giving again most of the weight to the to the part of the experiment we do not understand. Is that not an issue?

What is the objective of all the section 3.2? No particle phase chemistry was studied in the paper. Please, change the title 3.2. Most of the sentences in section 3.2 are generalities, figures are listed but poorly discussed and the selected figures do not justify the reached conclusions. The writing of this section has to be largely improved with clear objectives and appropriated figures, discussion of the results and comparisons with literature.

Why are the activity coefficients needed (p.11 l.20)?

In the two product parameterization, a1=a2 and Kom1=Kom2. What optimization method has been used? Does this mean that one product is enough to parameterized SOA formation? How can this be explained?

**Technical corrections**

- p.2 l.32: change "$\mu$ g m3" into "$\mu$ g m-3"

- p.3 l.9: I don't see a link between this study and the "theoretical foundation" you are talking about. I would remove "theoretical foundation".

-p.3 l.27: "formic acid was added to experiment 6 and 7 to ascertain the role of SCI". What do you mean by "ascertain the role of SCI"? Are you talking about the role of CI?

-p.7 l.9: change "acetaldhyde" into "acetaldehyde"

-p.8 l.15: this part of the sentence "So whilst the complete product distribution will likely consist of a myriad of species (Aumont et al., 2005)" is off-topic.

-p.36 Fig.10: in the legend, "$\mu$ g m-3" has to be changed in a concentration in volume.

-p.31 Fig.5: $\Delta$[OH] on the y axis?

- check that the concentrations and the variation of the concentration are written as [X]

and not X

---

## Referee Comment (RC2) · Anonymous Referee #2 · 31 Mar 2017

General Comments

In this manuscript the authors describe results of a series of environmental chamber experiments in which a-phellandrene was reacted with ozone under a variety of conditions. Gas phase products were monitored with a PTR-MS and aerosol composition with an AMS. Yields for a number of low molecular weight gas phase products, OH radicals, and SOA were measured, and the rate constants for the reaction of a-phellandrene and first generation products with ozone were determined. A large number of higher molecular weight gas phase products were also observed, but could not be identified. Aerosol elemental composition, carbon oxidation state, and density were also measured. Overall, these experiments were well done and generated a large volume of data on gas and aerosol composition. The data analysis is very thorough and the authors have extracted about as much from this study as possible. The manuscript

reports useful new information on an important monoterpene emission, which may be an important contributor to nucleation and SOA formation in some forested areas. When combined with results of ongoing molecular analysis of SOA products should provide a more complete picture of the chemistry. I think the manuscript is suitable for publication, but suggest the following comments be addressed.

Specific Comments

1. Page 5, lines 23–25; Page 6, lines 5–9: What role might gas-wall partitioning of products play in these observations? It is now well established that this can be important for oxidized compounds. The authors could explore the extent to which their products might partition to the walls using empirical approaches that have recently been developed (e.g., Krechmer et al. ES&T, 2016).

2. Page 6, 32–35: Are NO3 radicals formed under the conditions of these experiments, and if so how might that chemistry affect the results?

3. Page 7, lines 21–23: Are you certain that organic nitrates would be observed with the PTR-MS? It seems likely that these compounds could lose HNO3.

4. Page 10, lines 1–25: No mention is made of the possible effect of particle-phase reactions of O3 with first-generation products. These reactions would be very fast (reactive uptake coefficient about 0.001) and could form a variety of low volatility products.

5. Page 12, lines 26–27: Please provide a reference for the proposed relationship between density and phase state.

6. Page 12, lines 26–27: How well do the measured densities compare with those expected from measured O//C/H composition. Parameterizations have been developed for this (e.g. Kuwata et al. ES&T, 2011).

7. Page 13, lines 5–6: The reported wall loss rates for particles seem to be much higher than those measured by others, especially for such a large chamber. Any idea why?

8. Page 13, lines 25–26: I would have expected that for a four-parameter fit would give a set a four different values. Doesn't this result imply that a two-parameter fit is as good as a four-parameter fit?

9. Because of the large rate constant for reaction of a-phellandrene with O3 it seems likely that these experiments were conducted under conditions where RO2-RO2 reactions dominate the radical chemistry. In a clean atmosphere where this reaction might be thought to play a role in nucleation, it is likely that autoxidation may be important, or RO2-HO2 reactions. Some discussion of this would be useful.

Technical Comments

None.

---

## Author Comment (AC1) · 18 Apr 2017

**Response to referees for, "Ozonolysis of α-phellandrene, Part 1: Gas- and particle-phase characterisation" by Mackenzie-Rae et al.**

The authors would like to thank the referees for their time reviewing the manuscript, and for the thoughtful feedback provided. Based on their recommendations a number of significant modifications have been made to improve the manuscript, which we believe has substantially improved the interpretation of this study. Presented below are the specific comments made by the reviewers (italicised) and our corresponding responses (non-italicised).

**Referee #1**

*General comments*
*This paper presents experimental results on the formation of gaseous organic species, OH radicals and secondary organic aerosols (SOA) during a-phellandrene ozonolysis. Various dark ozonolysis experiments were performed in an indoor smog chamber facility (with or without OH scavengers, CI scavengers and NOx). Gas-phase species were monitored using a PTR-TOF-MS, SOA size distributions with a SMPS and aerosol compositions with an AMS. Measurements were used to calculate yields for a few gaseous organic species, OH yields from a-phellandrene and its first-generation products, and rate coefficient of first-generation products. Aerosol yields and effective density of SOA are also provided. The reactivity of a-phellandrene with ozone, as well as its impact on HOx and SOA formation has currently not been studied in the scientific community. This paper provides experimental data that are valuable to better understand the environmental impact of biogenic compound ozonolysis and could therefore be of interest to the scientific community. However, the manuscript suffers from a lack of clarity, precision and discussion in several places. In particular, the current knowledge on a-phellandrene ozonolysis is not enough presented, the purpose of the selected type of experiments are not explained, the methodology used to calculate yields or kinetic constants should be clarified and conclusions reached by the author should be better justified by comparison with previous work. All of this make that even if the results may well be fully valid, an explanation of the methodologies and a justification of conclusions are needed to support the results. Major revisions are therefore required before publication in ACP.*

> We appreciate the referee's feedback and their recognition of the value that the study offers the scientific community. We will now address their specific concerns to improve the precision, clarity and discussion of the manuscript.

*Specific comments*
*The chemical reactions expected to occur during a-phellandrene ozonolysis according to the literature are never described rigorously in the manuscript. The discussion and the Figure in S4 should be presented before section 2 (and not in section 3 where it is difficult to make the difference between our knowledge, the coherence of the observations shown in this study and the novelty of the results). This figure should be used in section 3 to justify the detection of some species and could be coupled with in Fig. 7 to explain the formation of the "new" species*

*detected in this study and with Fig. 6. The structure of a-phellandrene can also be presented from this figure, allowing to remove Figure 1 which is not really useful.*

Upon review it is clear that both the general ozonolysis mechanism, and the specific processes involved in α-phellandrene's degradation are not included or are included too late in the manuscript to prove efficient. To address the lack of background mechanism information, the following paragraph was added to the introduction.

p.2 l.20: "Ozonolysis is generally agreed to occur through a concerted cycloaddition of ozone to the olefin bond, forming a 1,2,3-trioxolane intermediate species referred to as a primary ozonide (POZ) (Calvert et al., 2000; Johnson and Marston, 2008). Addition of ozone is highly exothermic with excess energy retained in the POZ structure, resulting in rapid decomposition through homolytic cleavage of the C–C and one of the O–O bonds, which forms, in the case of asymmetrically substituted alkenes, a pair of products containing a carbonyl and reactive Criegee Intermediate (CI). Sufficient vibrational and rotational excitation exists in the CI to permit further unimolecular decomposition which typically occurs through one of two channels; firstly excited CIs can cyclise to a dioxirane, which then decomposes to a carboxylic acid, ester or lactone depending on neighbouring substituents in what is known as the ester or `hot' acid channel, or secondly, when available, excited CIs can isomerise via a 1,5-hydrogen shift to form a vinylhydroperoxide, which subsequently decomposes into a vinoxy radical and a hydroxyl radical in what is known as the hydroperoxide channel. Alternatively excited CIs can be collisionally stabilised, such that bimolecular reactions with trace species (e.g. $H_2O$, $NO_2$, aldehydes, acids) becomes important (Johnson and Marston, 2008). The relative prevalence of these two channels is strongly linked to the structure and conformation of the CI (Vereecken et al., 2015), with the various mechanistic pathways summarised in Fig. 1."

Furthermore the following sentence was added on p.3 l.8. to introduce the reactions expected to occur during α-phellandrene's ozonolysis before section 2.

p.3 l.8: "A basic overview of the reaction pathways is provided in Fig. 1, with a comprehensive discussion of the reaction mechanism of α-phellandrene with ozone based on the findings of Mackenzie-Rae et al. (2016) pertinent to this study provided in the Supplementary Information S.1."

Rather than discarding, Figure 1 was amended (shown below) to include an overview of processes involved in the degradation of α-phellandrene through ozonolysis. The revised figure is accompanied by direction in the main text to the Supplementary information S1 (previously S4), for those readers seeking a more exhaustive description of the mechanistic processes involved in α-phellandrene's degradation through ozonolysis. Furthermore the coupling between the various mechanistic figures was

improved to assist in overall readability. The lines starting at p.5 l.22 were amended to:

p.5 l.22: "Ignoring conformational isomerism, the ozonolysis of α-phellandrene can yield four unique CIs (Fig. 1) (Mackenzie-Rae et al., 2016), with the degradation mechanism of **CI3** provided in Fig. 7. Detailed schematics of the remaining CIs are provided in the Supplementary Information (S1), and lead to products isomeric to those shown in Fig. 7." The caption of Figure 7 was then changed to:

Fig. 7: "Partial mechanism for the ozonolysis of α-phellandrene starting from **CI3**, yielding product masses detected by the PTR-TOF. Similar constructs for the remaining CIs are provided in the Supplementary Information (S.1)."

Similarly the caption of Figure S.1.1 (formally S.4.1) was adjusted to link into the degradation scheme provided in the main text, by adding the following sentence to the end of the caption:

Fig. S.1.1: "A more exhaustive description of the mechanism originating from **CI3** is provided in Fig. 7 of the main text."

A rigorous description of the mechanism is not the goal of this paper, and is more or less covered in the theoretical paper, 'Computational investigation into the gas-phase ozonolysis of the conjugated monoterpene α-phellandrene' by Mackenzie-Rae et al. (2016). However with the updates we now think that the introduction provides the reader with the necessary background knowledge to understand the mechanistic aspects and discussion provided in Section 3, and improves the overall clarity of the manuscript.

[Figure]

Figure 1. Simplified mechanism showing reaction processes involved during ozone addition to α-phellandrene within conventional frameworks (adapted from Mackenzie-Rae et al. (2016)). Carbon labels on α-phellandrene are referred to in the main text.

*The various experiments are listed but the objective of each different type as well as*

*the expected impact on the chemical system should be discussed (why several O3 injections, NO2 addition? what is the expected impact of an OH and a Criegee intermediate (CI) scavenger on the chemistry? how much of the CI is expected to be scavenged by the used amount of formic acid?). The observed influences of OH and CI scavengers, NO2 or the several O3 additions are not enough shown or discussed, questioning the interest of including exp. 6, 7, 9 and 11 in the paper. Do the authors see expected or unexpected differences in gaseous secondary organic and OH yields, SOA formation...?*

We thank the referee for noticing these points; justification of the experimental design and their consequences were not sufficiently addressed in the original manuscript. With respect to the several additions of ozone in some experiments (it was only ever one additional addition) the following lines were added/amended:

p.3 l.25: "α-phellandrene was injected prior to admission of $O_3$ into the chamber, with $O_3$ added through two separate additions in experiments 7 and 10 to facilitate the identification of detected species as either first- or second-generation products."

p.5 l.34: "This continual increase remained true in experiments which added a large secondary dose of ozone after commencement of the reaction (Fig. S.4.1), confirming the ions discussed as saturated."

With respect to $NO_2$ addition the following lines were added/amended:

p.3 l.29: "Prior to $O_3$ addition in experiment 11, 385 ppb of $NO_2$ was added through a septum installed in one of the injection ports using a gas-tight syringe, with the inclusion providing an alternative representation of tropospheric nocturnal chemistry in a polluted environment."

p.7 l.10: "A comparison of rate constants of $O_3$ with α-phellandrene (3.0 x $10^{-15}$ $cm^3$ $molecule^{-1}$ $s^{-1}$) and $NO_2$ (3.5 x $10^{-17}$ $cm^3$ $molecule^{-1}$ $s^{-1}$) suggests that the majority of $O_3$ will be consumed by α-phellandrene, with formation of the nitrate radical relatively minor. Nevertheless $NO_2$ is in excess in the system, with a systematic reduction in product yields indicative of a shift towards $RO_2+NO_2$ chemistry, producing peroxy nitrate containing products (Draper et al., 2015)."

Reference Added:
Draper, D. C., Farmer, D. K., Desyaterik, Y., and Fry, J. L.: A qualitative comparison of secondary organic aerosol yields and composition from ozonolysis of monoterpenes at varying concentrations of $NO_2$, Atmos. Chem. Phys., 15, 12 267-12 281, 2015.

p.7 l.21: "...while addition of $NO_2$ to the system in experiment 11 resulted in significantly reduced yields, although overall distribution remained similar with no new peaks or evidence of nitrate containing compounds observed, indicating that ozonolysis products remain dominant."

p.12 l.23: "Similarly there is a reduction in the α-phellandrene normalised number distribution when $NO_2$ is added (Fig. 11). Like formic acid, $NO_2$ can also react with sCIs (Johnson and Marston, 2008) and therefore potentially inhibit particle formation and growth. If this were the case then results from this ozonolysis study likely represent an upper limit to SOA formation under ambient conditions, although more experiments are necessary to confirm the impact of $NO_2$ on SOA formation in the α-phellandrene system."

In addition Fig. 11a was amended to show the impact that $NO_2$ has with respect to the other experiments.

[Figure]

p.15 l.23: "Nitrogen containing species were found to make little contribution to the aerosol formed in experiment 11, with an average N/C ≈ 0.002 during this experiment. Nitrate functionality is believed to significantly reduce the vapour pressure of constituents (Capouet and Müller, 2006; Pankow and Asher, 2008), with the result implying a small gas-phase concentration. Nevertheless there exists evidence that organic nitrate contribution to SOA may be kinetically, rather than volatility driven (Perraud et al., 2012)"

Reference Added:
Perraud, V., Bruns, E. A., Ezell, M. J., Johnson, S. N., Yu, Y., Alexander, M. L., Zelenyuk, A., Imre, D., Chang, W. L., Dabdub, D., Pankow, J. F., and Finlayson-Pitts, B. J.: Nonequilibrium atmospheric secondary organic aerosol formation and growth, Proc. Natl. Acad. Sci., 109, 2836-2841, 2012.

With respect to formic acid addition it is difficult to estimate the amount of sCI that is expected to be scavenged, as this would require knowledge of sCI formation rates and rate constants of the various competing pathways (bimolecular reactions, unimolecular decomposition, reaction with formic acid), none of which currently exists in the literature for α-phellandrene. For this reason the information is not given, however the amount of formic acid has now been added to Table 1, with the ratio of formic acid to α-phellandrene added comparable to similar studies where

it has been used as a sCI scavenger (e.g. Bonn et al., 2002; Winterhalter et al., 2009). The following lines were also added/amended to strengthen the argument for its inclusion and discussion of its impact:

p.3 l.27: "Formic acid (J&K Scientific Ltd., 98%) was added to experiments 6 and 7 as a stabilised Criegee Intermediate (sCI) scavenger to better understand the role of sCIs on gas-phase species distribution and importantly particle-phase formation and growth, for which it has been identified as a significant precursor (Bonn et al., 2002; Bateman et al., 2009; Sakamoto et al., 2013; Wang et al., 2016)."

p.7 l.8: "The addition of a sCI scavenger was found to have little impact on product distribution or yields, suggesting that the sCI-formic acid complexes ultimately decompose to yield similar gas-phase products as sCIs that degrade through conventional channels. Whether this decoupling of the complex occurs inside the reactor or upon protonation in the PTR-TOF remains unknown."

The addition of cyclohexane as an OH scavenger in ozonolysis experiments is fairly common practice in the field (e.g. Bonn et al., 2002; Keywood et al., 2004; Saathoff et al., 2009; Winterhalter et al., 2009) and so validation of its inclusion is not necessary. However with regards to experiment 9 it is rightly noted by the referee that validation of its inclusion and impact on the gas-phase were not included, with the following lines added/amended to rectify this.

p.3 l.25: "Anhydrous cyclohexane (Sigma-Aldrich, 99.5%) was added in sufficient quantity in all but two experiments to scavenge > 95% of OH radicals (Aschmann et al., 1996; Herrmann et al., 2010), with the remaining experiments used to assess the impact of cyclohexane's inclusion."

p.7 l.9: "Similarly no significant differences in yields were observed between experiment 9 and OH-scavenged experiments, with decomposition into smaller carbon species counter intuitively invariant to action by the OH radical; strengthening the argument that fragmentation inside the PTR-TOF is non-negligible"

p.7 l.21: "Again the presence of OH radicals in experiment 9 had little effect on product yields…"

*What is the typical volume of the chamber at the end of an experiment? (p.3 l.31).*

Varies for each experiment, but typically between 6 – 8 $m^3$. This information was added to the end of p.3 l.31.

*Numerous figures and tables are discussed before or without being presented. Each Figure or Table should be clearly introduced in the text before being discussed.*

Yes, upon re-reading we agree that the integration of figures/tables into text definitely requires improvement in multiple places. The following changes were made to address this:

The premature reference to Fig. 2 on p.5 l.6 was removed, with the proper introduction to the figure remaining at p.5 l.20.

Figures 3, 4, 5, 6 and 7 were renumbered to Figs 4, 6, 7, 5 and 3 respectively, so that the order of introduction in the text matches the figure number. This makes logical sense in the scheme of the manuscript and ensures figures have properly been introduced before discussion, improving general readability.

Additionally the following sentences were changed to better introduce the figures.

p.8 l.6: "An example showing this from a proposed first-generation product is provided in Fig. 5."

p.11 l.28: "Nevertheless rapid aerosol formation is observed upon reaction of α-phellandrene and ozone as shown in Fig. 10, with sharp increases in particle number (dN/dlogDp) and volume (dV/dlogDp) concentrations observed."

p.15 l.16: "Figure 16 shows the typical temporal profile of aerosol composition observed over an experiment."

p.16 l.27: "The carbon mass balance for each experiment is shown in Fig. 19. It was calculated by…"

The lines p.4 l.30-33 that introduce Table 1 have been moved to p.3 l.31, with the earlier reference to the table on p.3 l.24 removed.

Reference to Table 2 on p.5 l.9 was removed, as it was unnecessary and occurred before Table 2 was properly introduced on line 18 of that page.

Table 3 is now properly introduced on p.6 l.18 by adding the following, with the old reference on p.6 l.28 removed.

p.6 l.18: "The average yield from sequential ozonolysis is therefore calculated with results provided in Table 3. In practice however calculations are…"

*The temporal evolution of the PTR-TOF-MS signals presented in Fig 2 should be described in the text at the beginning of the section 3.1.1. In particular, Why are all concentrations at around 35 min going to 0 except m/z = 42 and 137? Why does the precursor's concentration decrease at the same time? The m/z 42 signal is not impacted. Why? Is the linear decrease of O3 (after 70 min) due to wall loss?*

Yes Fig. 2 was not discussed in depth in the text. This was partly because it shows time profiles from one of eleven experiments, nonetheless as the referee pointed out there are some common trends which are worth discussing. For this reasons the line starting at p.5 l.20 was expanded to the following.

p.5 l.20: "Figure 2 shows time profiles of major species detected by the PTR-TOF during the ozonolysis of α-phellandrene. For clarity peaks have been corrected for background readings recorded prior to the introduction of ozone. Upon the injection of ozone, α-phellandrene is rapidly oxidised forming a number of product ions at low concentrations that continually increase throughout the experiment. Meanwhile ozone, after rapid initial consumption, slowly decreases throughout the experiment in part due to losses to the reactor walls (Wang et al., 2014). The stability of acetonitrile and cyclohexane signals supports the finding of Wang et al. (2014) that wall losses are relatively minor for volatile organics in the GIG-CAS chamber."

With regards to the precursors concentration decreasing at the start and the corresponding m/z 42 signal remaining unchanged, this was an irregularity experienced in this experiment caused by the fans being switched on after α-phellandrene's injection. As the sampling port is near to the injection port, upon addition of α-phellandrene its concentration was read erroneously high. When switched on the fans quickly dispersed α-phellandrene causing its concentration to fall and stabilise. As the fans were switched on prior to acetonitrile injection such an effect was not seen. Justification of this was thus added to the caption of Fig 2.

Fig. 2: "The peak of α-phellandrene observed upon its addition was the result of the reactor fans being switched on immediately prior to the introduction of acetonitrile in this experiment."

With regards to impact all concentrations and physical conditions (e.g. temperature, humidity) were stable prior to injection of ozone, so the delayed start of the fans in this experiment had no impact on findings.

*In OH scavenged experiment, the reactivity of the system is expected to stop when the two a-phellandrene double bonds have each been reacting with O3. PTR-TOF-MS measurements show an increase of all the m/z signals (also for small molecules), except for the m/z assigned to the precursor and the inert gas. Some of these m/z could be assigned to first generation products (and also maybe second generation?) expected to be formed during scavenged a-phellandrene + O3 experiments. These first and also second generation species are expected to stop growing when no double bond remain in the molecule considering the known chemistry. The authors claim at several places in the paper that the observed increase suggests that these compounds are second-generation or higher generation species (e.g. p.5 l.25, l.35 and below in the paper) or that this unique profile implies that they derived from a source secondary to ozonolysis, such as gas-phase accretion (e.g. p7 l.35., table 2, S3...). The authors should be careful in*

*drawing their conclusions. Why an increasing signal makes the species a second-generation compound?*

In the majority of experiments ozone is added in excess ($[O_3]$ = 2x[α-phellandrene]). Therefore first-generation products, containing one residual double bond, are expected to further react resulting in a time-profile that reaches a peak sometime after initiation, before decaying as the first-generation product is consumed by ozone (e.g. Lee et al., 2006; Ng et al., 2006; Camredon et al., 2010). Because such time profiles were not observed it can be concluded that either (i) all observed species have second-generation contribution or (ii) experimental run times were not long enough to see the consumption of first-generation products. The simple modelling study described in the manuscript shows (ii) not to be true. The same thought process, whereby increasing signals were argued to have second-generation contributions was made in Lee et al. (2006). This distinction was not made clear on p.5 l.25 and l.33, with the following changes made:

p.5 l.25: "…however none of the product ions detected were observed to decrease over the course of the chamber experiments, suggesting that detected ions in part correspond to second-generation products."

p.5 l.33: "Both these ions were detected in the PTR-TOF but again had concentrations which increased throughout the experiments, suggesting that they have large contributions from saturated species."

The study of Ng et al. (2006), whilst referenced in the manuscript, was not used as an additional example of a study where first-generation products were observed with a PTR-MS for all terpenes investigated except for the most reactive poly-alkenes (in their case α-terpinene, α-humulene and β-caryophyllene). The reference was thus incorporated into p.6 l.1 alongside that of Lee et al. (2006).

*If this increasing signal comes from the formation of dimers, why is an increase also observed for the low m/z signals (m/z = 47, 59, 61…)?*

An increasing signal does not necessarily mean it comes from dimers, we argue that it purely means it likely comes from a saturated species. We hope that the above changes make this clearer. Instead dimers are used to explain the higher mass signals (*m/z* 167, 169 and 185) as due to fragmentation upon the second addition of ozone products of these masses are unlikely to form in large amounts (requires ~5 oxygens on a $C_7$ or less backbone). Furthermore as Fig. S.3.1 shows the rate of formation of these products is relatively unperturbed by a second addition of ozone, whilst direct second-generation products (e.g. *m/z* 47, 59, 61) all show a marked increase. Because of this a higher generation process, such as accretion, is inferred. P.7 l.32 and the supplementary entry S.3 were changed to better convey this.

p.7 l.32: "...have relatively constant temporal profiles which also lack an accelerated increase upon a second addition of ozone; a feature that is apparent among lighter product ions (Fig. S. 3. 1). Their unique time profiles imply that they are derived from a source secondary to ozonolysis such as gas-phase accretion reactions, with modeling support for this provided in the Supplementary Information (S.3).

S.3: "Signals at *m/z* 167, 169 and 185 are relatively invariant to a second addition of ozone to the reactor, lacking the characteristic rapid increase in concentration observed for lighter product ions (Figure S.3.1). This suggests that the peaks are formed through a process supplementary to direct ozonolysis."

*The hypothesis of some chamber wall artefacts is never discussed in the paper. Have the author already tested the impact of chamber teflon walls on the off-gassing of radical and/or organic species in the gas phase? Could it be a possible explanation of the increasing m/z signals?*

It is not mentioned in the paper but off-gassing of radicals/organics from the reactor walls is not expected to be important under the dark, dry conditions used, especially given the extensive cleaning process between experiments (Section 2.1). Characterisation of the chamber auxiliary process is described in depth in Wang et al. (2014), with off-gassing expected to be predominantly a light driven process. Wang et al. (2014) showed that propene remains stable in concentration, whilst we conducted a similar experiment with α-phellandrene whose concentration was also stable inside a clean reactor, suggesting that off-gassing of radicals is minor. This is somewhat shown in Fig. 2, where both α-phellandrene and product signals remain stable in the ~30 minutes prior to ozone addition. To address this in the manuscript the following line was added.

p.3 l.19: "...with the impact of off-gassing of radicals from the reactor walls during experiments under the dark, dry conditions used negligible (Wang et al., 2014)."

*Experimental results are used to provide various yields. The yields are however not always clearly defined in the paper. For secondary organic species (SOC), Y would be usually defined as: Y = Δ [SOC]/ Δ [COV]. Here Y is calculated from the amount of O3 that has reacted as Y = Δ [SOC]/ Δ [O3]. Has the Δ [O3] to be divided by a factor of 2 to correspond to the Δ [COV] or not? Has this method been used previously? How do you deal with experiments without OH scavenger or with various additions of O3? The initial concentration of O3 is not known. Does this have an impact on the calculated yields? Looking at Fig. 3, no data is used at the beginning of the experiment when the system is the most reactive, i.e. before Δ [O3]=40 ppb. Why? The intercept of the regression lines is different of 0. Why? The all length of the experiment is used to optimize regression lines and therefore to get the yield. 4/5 of these measured data correspond to a decrease of O3 concentration due to wall loss and an increase of secondary organic species which is difficult to understand. Is*

*that not an issue to keep these points for the optimization?*

We assume that the referee is defining SOC as secondary organic compound and means VOC instead of COV? If so, then the yield equation provided is correct for determining yields with respect to α-phellandrene (and was used for OH yields in this work). However $\Delta[O_3]$ was used as opposed to $\Delta[VOC]$ in calculating gas-phase product yields in this study because of the high reactivity of α-phellandrene. For all species, most of the growth occurs after α-phellandrene is more or less completely consumed; therefore if the primary hydrocarbon were used in the denominator unrealistically high yields would be obtained. The use of $\Delta[O_3]$ provided much cleaner, more consistent results – however as far as the authors are aware it is the first time this method has been used. The implication of using $\Delta[O_3]$ is stated on p.6 l.18, 'The average yield from sequential ozonolysis is therefore calculated'. This statement implies $\Delta[O_3]$ was not divided by a factor of 2, this is because this action would implicitly assume that half of each product yield is attributable to α-phellandrene. Obviously this is going to be different for each product ion, and so making this assumption would be incorrect. The wall loss of ozone was frequently calibrated and corrected for in all calculations (p.6 l.17). The fact that 4/5 of the data occurs after α-phellandrene's consumption is an issue for optimisation as it results in fitted yield plots being dominated by reaction of first-generation products, however this shortcoming is noted on p.6 l.19. The difference in methodologies when comparing results is now noted on p.6 l.30.

p.6 l.30: "...although a subtle difference in methodology should be noted with Lee et al. (2006) calculating yields with respect to the parent hydrocarbon."

For experiments with two additions of $O_3$ separate yield lines were fitted, one for each addition. Differences were observed, as the yield plot from the first addition contains a much greater influence from α-phellandrene, whilst the second addition is impacted to a greater extent by first-generation products. However in keeping with the remaining data the two yields were averaged to get an overall yield from the system. This subtle change in methodology is now included in p.6 l.23.

p.6 l.23: "For experiments that had two additions of ozone (7 and 10), separate yield lines were fitted for data after each addition of ozone with the results then averaged, therefore maintaining the reported yield as an average of the entire ozonolysis system."

For the two experiments where no OH radical scavenger was added no change in analysis methodology was made, with the impact of OH radicals an obvious unaccounted for uncertainty. This impact is implicit in the new inclusions of p.7 l.9 and p.7 l.21 described on page 6 of this document.

Whilst it is true that the initial concentration of $O_3$ is not known it has no

impact on results, as it is the change in [$O_3$] between data points that is important. With regards to Fig. 3, it is well noted that no data points are used when the system is at its most reactive ($\Delta$[$O_3$] < 40 ppb). This is because of the finite mixing time of the reactor, with the first few minutes of data not used to ensure [$O_3$] readings are accurate (mentioned on p.6 l.20). For the plotted experiment (#5), this resulted in a difference of 38 ppb between the peak [$O_3$] reading and first data point used. The intercepts are non-zero because of this; with the majority of data points used corresponding to reaction of first-generation products. The impact of excluding data from the first few minutes, predominantly involving product production from α-phellandrene, ultimately yields non-zero intercepts. This was observed across all experiments.

*Two OH yields are calculated, one for a-phellandrene and another one for its first-generation products using the methodology presented in Herrmann et al. (2010). The methodology used here should be presented using the Herrmann et al. (2010) reference with the two reactions p.8 l.34, before the presentation of the OH yield results.*

We agree this change will improve the flow of the manuscript. The sentence referring to the results on p.8 l.18 is therefore moved to p.9 l.12, so that results are presented after the method.

*How do you define the end of the a-phellandrene dominated zone (Fig. 4)? Why is the intercept of the regression line different of 0 in Fig 5.a.? The intercept could be 0 with a good correlation coefficient if the end of the a-phellandrene dominated zone stops before.*

The end of the α-phellandrene dominated zone for each experiment is chosen qualitatively, however a correlation coefficient ($R^2$) greater than 0.9 is always maintained. The intercept of the regression line in Fig. 5a is different from zero due to a delay in OH radical detection by the PTR-TOF. There appears to be little change in OH radical production over the first 3 data points, corresponding to a 6 second delay with respect to α-phellandrene consumption. OH radicals do have to react with cyclohexane to form cyclohexanone to be detected, although 6 seconds to do this does seem too long given the rapid reactions of radicals. Nevertheless similar delay lengths were observed in all experiments.

*How do you define the end of the product dominated zone (Fig. 4)? Is that not when the observations are difficult to understand and when we do not look at ozonolysis?*

The product dominated zone had no end-point. We propose in theory it would be when the products are no longer reacting, being when all species are saturated in an ozonolysis experiment, although this point was never reached during chamber simulations considered here. For example Fig. 5b shows all data points until termination of experiment 3. It is not defined as a point when "observations are difficult to understand".

*In Fig 5.b, shouldn't the OH and O3 concentration variations be looked starting from the beginning of the product dominated zone (and not from the beginning of the experiment) and why is the intercept of the regression line far from 0?*

In Fig. 5b the variation of OH and $O_3$ are looked at from the start of the product-dominated regime. That is why both [OH] and [$O_3$] are non-zero. The intercept of the regression line for this is far from zero because it does start from the beginning of the product-dominated regime, with [OH] therefore having received contributions from α-phellandrene. As the OH yield from α-phellandrene is higher than the yield from first-generation products (Herrmann et al. 2010), a positive y-intercept is produced upon fitting the data.

*The kinetic constant for the reaction of the secondary products with O3 was optimized giving again most of the weight to the to the part of the experiment we do not understand. Is that not an issue?*

The beauty with the modeling study is that due to its simplicity, all parameters have to be more or less correct to get a time-profile that matches experiments. Whilst the majority of data pertains to the end of the experiment, the constants concerning the reaction of α-phellandrene with ozone are extremely important in determining overall shape and magnitude of the profiles. Indeed these parameters were found to be extremely sensitive and more difficult to fit. Therefore the quantity of data in the latter part of the experiment is not an issue.

*What is the objective of all the section 3.2? No particle phase chemistry was studied in the paper. Please, change the title 3.2. Most of the sentences in section 3.2 are generalities, figures are listed but poorly discussed and the selected figures do not justify the reached conclusions. The writing of this section has to be largely improved with clear objectives and appropriated figures, discussion of the results and comparisons with literature.*

Correct no particle-phase chemistry is discussed in Section 3.2, with this aspect reserved for the companion paper. Section 3.2 is thus renamed, 'Particle-phase Analysis', inline with the naming of Section 3.1, 'Gas-phase Analysis'.

A large portion of the generality discussion occurs when using metrics to assess the particle-phase. It is true that these parts are very qualitative, offering mere insights with strong conclusions not reached. For this reason the text from p.14 l.31 – p.15 l.11, discussing the $CO_2^+$ to $C_2H_3O^+$ ratios, and text on p.16 l.1 – p.16 l.12, discussing the double bond index, are removed from the discussion. This includes removal of Fig. 15 and the line corresponding to it in the abstract p.1 l.18. In total 29 lines were removed with no major implication on the overall discussion, making Section 3.2 much more succinct with clear objectives.

The introduction of the AMS data in the discussion was also heavily

modified, with large portions removed, including reference to and inclusion of Fig. 14. The paragraph starting at p.14 l.21 has been replaced with:

p.14 l.21: "Resolution in the W-mode of the AMS is sufficient to unambiguously identify chemical formulae of detected ions (DeCarlo et al., 2006; Aiken et al., 2007). Ions are formed however using high-energy electron impact ionisation (70 eV), resulting in significant fragmentation. The complexity of aerosol produced, along with an unknown number of fragmentation pathways including the possibility of charge migration and other internal rearrangements, makes it exceedingly difficult to obtain clear structural information about SOA constituents from the AMS. For this reason filter samples were collected and analysed to identify SOA constituents, with results to be published in a companion paper. Nevertheless the AMS remains useful for analysing bulk properties of the aerosol to gain further insight into the system."

Furthermore minor improvements were made to the remaining text to improve clarity and the overall discussion.

p.12 l.17: To improve flow and readability the text, 'whether uni- or bi-molecular' was removed.

p.13 l.27: Further references added to show utility of result.
Tsigaridis, K. and Kanakidou, K.: Global modeling of secondary organic aerosol in the troposphere: a sensitivity analysis, Atmos. Chem. Phys., 3, 1849-1869, 2003.
Henze, D. K. and Seinfeld, J. H.: Global secondary organic aerosol from isoprene oxidation, Geophys. Res. Lett., 33, L09812, 2006.
Jathar, S. H., Cappa, C. D., Wexler, A. S., Seinfeld, J. H., and Kleeman, M. J.: Simulating secondary organic aerosol in a regional air quality model using the statistical oxidation model – Part 1: Assessing the influence of constrained multi-generational ageing. Atmos. Chem. Phys., 16, 2309-2322, 2016.

p.13 l.31: Line added, "Nevertheless yields from the two experiments differ by almost a factor of two despite having similar starting conditions, with further experiments necessary to better quantify the impact of sCIs on yields."

p.15 l.29: To make reference to the literature stronger, "Figure 17 shows that the $OS_c$ decreases from -0.61 to -1.00 as the particle loading increases from 21.5 to 658.1 $\mu g\ m^{-3}$, suggesting a strong link between mass loading and degree of functionalisation consistent with the findings of Shilling et al. (2009) for the ozononlysis of $\alpha$-pinene."

p.16 l.26: References added:
Turpin, B. J., Saxena, P., and Andrews, A.: Measuring and simulating particulate organics in the atmosphere: Problems and prospects,

Atmos. Environ., 34, 2983-3013, 2000.

Kirchsetter, T. W., Corrigan, C. E., and Novakov, T.: Laboratory and field investigation of the adsorption of gaseous organic compounds onto quartz filters, Atmos. Environ., 35, 1663-1671, 2001.

*Why are the activity coefficients needed (p.11 l.20)?*

Activity coefficients affect the vapour pressure of SVOCs in the condensed organic phase (Pankow, 1994), with activity coefficient estimations therefore required for calculating saturation vapour concentrations. From the theory of Pankow (1994):

$$p_i = X_{i,om}\zeta_i p^0_{L,i}$$

where $p_i$ is the gas-phase pressure of compound i, $X_{i,om}$ is its mole fraction in the organic matter phase and $\zeta_i$ is its activity in the organic matter phase.

*In the two product parameterization, a1=a2 and Kom1=Kom2. What optimization method has been used? Does this mean that one product is enough to parameterized SOA formation? How can this be explained?*

The parameters in Equation 2 in the manuscript were optimised for a two-product model using the NonlinearModelFit function in Wolfram Mathematica software, constraining each variable to be positive. Corresponding fitted parameters did not differ within the first five decimal points and so were reported as the same. A two-product model was reported, as is convention in the field (Odum et al., 1996). However because the parameters are the same it does result in a one-product model producing an identical fit:

$$Y = \Delta M_o \left( \frac{0.60 \times 0.022}{1 + 0.022\,\Delta M_o} + \frac{0.60 \times 0.022}{1 + 0.022\,\Delta M_o} \right) = 2\Delta M_o \frac{0.60 \times 0.022}{1 + 0.022\,\Delta M_o} = \Delta M_o \frac{1.2 \times 0.022}{1 + 0.022\,\Delta M_o}$$

where $\alpha_1 = 1.2$ and $K_{om,1} = 0.022$. Upon the referees recommendation the manuscript was changed to reflect this, with a more economical one-product fit now included.

*Technical corrections*
*- p.2 l.32: change "µ g m3" into "µ g m-3"*

Thank you, it has been corrected.

*- p.3 l.9: I don't see a link between this study and the "theoretical foundation" you are talking about. I would remove "theoretical foundation".*

The line, 'With a theoretical foundation', has been removed.

*-p.3 l.27: "formic acid was added to experiment 6 and 7 to ascertain the role of SCI". What do you mean by "ascertain the role of SCI"? Are you talking about the role of CI?*

Yes, the impact of stabilised Criegee intermediates on the reaction

mechanism and subsequent gas- and particle-phase observations. This line has been amended to improve clarity, with the revision provided on page 5 of this document.

*-p.7 l.9: change "acetaldhyde" into "acetaldehyde"*

Thank you, it has been corrected.

*-p.8 l.15: this part of the sentence "So whilst the complete product distribution will likely consist of a myriad of species (Aumont et al., 2005)" is off-topic.*

Agreed, it has been removed.

*-p.36 Fig.10: in the legend, "µg m-3" has to be changed in a concentration in volume.*

Thank-you, has been changed to $\mu m^3 \, m^{-3}$.

*-p.31 Fig.5: Δ[OH] on the y axis?*

[OH] starts at zero in the reactor, so [OH] = Δ[OH]. However the change was made for clarity, and for consistency with Fig. 4 where it was used.

*- Check that the concentrations and the variation of the concentration are written as [X] and not X .*

Amendments were made to Fig. 2 legend, Fig. 3 axis and legend, Fig. 4 axis, Fig. 5 axis and legend and Fig. 8 axis and legend.

Whilst strictly $M_o$ and HC are concentrations on p.14 and so should be written as $[M_o]$ and [HC], they are presented in the form standard in the literature and so brackets are not included (e.g. Odum et al., 1996; Griffin et al., 1999; Kroll et al., 2008). This additionally extends to the y-axis label in Fig. 13.

**Referee #2**

*General Comments*
*In this manuscript the authors describe results of a series of environmental chamber experiments in which a-phellandrene was reacted with ozone under a variety of conditions. Gas phase products were monitored with a PTR-MS and aerosol composition with an AMS. Yields for a number of low molecular weight gas phase products, OH radicals, and SOA were measured, and the rate constants for the reaction of a-phellandrene and first generation products with ozone were determined. A large number of higher molecular weight gas phase products were also observed, but could not be identified. Aerosol elemental composition, carbon oxidation state, and density were also measured. Overall, these experiments were well done and generated a large volume of data on gas and aerosol composition. The data analysis is very thorough and the authors have extracted about as much from this study as possible. The manuscript reports useful new information on an important monoterpene emission, which may be an important contributor to nucleation and SOA formation in some forested areas. When combined with results of ongoing molecular analysis of SOA products should provide a more complete picture of the chemistry. I think the manuscript is suitable for publication, but suggest the following comments be addressed.*

Thank-you. We will now address the specific comments.

*Specific Comments*
*1. Page 5, lines 23–25; Page 6, lines 5–9: What role might gas-wall partitioning of products play in these observations? It is now well established that this can be important for oxidized compounds. The authors could explore the extent to which their products might partition to the walls using empirical approaches that have recently been developed (e.g., Krechmer et al. ES&T, 2016).*

The loss of volatile organics to the reactor walls was addressed in a response to the first referee on page 7. However rightly pointed out, a substantial discussion of the impact of gas-wall partitioning in the system is missing from the original manuscript. The extent to which gas-wall portioning occurs depends heavily on the equivalent organic mass concentration of the Teflon walls ($C_w$), which has not been measured for the GIG-CAS reactor. Values for $C_w$ vary considerably in the literature, and so no formal parameterisation of the effect was undertaken. Nevertheless based on recent literature, it is likely that gas-wall partitioning of oxidised organics has a large impact on the results of the study, with the following updates and amendments made to the original manuscript to reflect this.

p.6 l.7: "Recent literature has shown that functionalised organic species experience considerable losses to Teflon chamber walls through gas-wall partitioning (e.g., Matsunaga and Ziemann, 2010; Zhang et al., 2014; Yeh and Ziemann, 2015; Krechmer et al., 2016; La et al., 2016). Observations indicate that organic compounds are not lost to the reactor walls, but rather partition between the gas-phase and Teflon walls in a reversible process that eventually reaches equilibrium, the speed of which is

dependent on reactor geometry, turbulence and species diffusivity, and penetration and accommodation in the reactor walls. Based on the work of Krechmer et al. (2016) the time scale for reaching gas-wall equilibrium in these experiments is thought to be less than 600 s. Gas-wall partitioning therefore operates quick enough to affect the considered chamber experiments and detection of first-generation products. The relative impact of gaseous wall losses is further explored in Section 3.2.1, nonetheless partitioning is strongly dependent on volatility with losses of highly-functionalised first-generation products of α-phellandrene to reactor walls and/or sample lines during transfer into and detection by the PTR-TOF expected (Yeh and Ziemann, 2015; Krechmer et al., 2016; La et al., 2016)."

p.7 l.21: "Calculated yields for these larger products were in general < 5%, with detected products sufficiently volatile such that gas-wall partitioning losses are thought to be minor (see Fig. 9)."

p.11 l.23: "Gas-particle partitioning occurs in competition with gas-wall partitioning, a process that is also dependent on species saturation vapour concentrations (Supplementary Information S.6). In parameterising gas-wall partitioning, the Teflon film is often considered to have an equivalent organic aerosol mass concentration ($C_w$). Values for $C_w$ vary significantly in the literature, with Ziemann and co-workers reporting values of $C_w \sim 2 - 40$ mg m$^{-3}$ (Matsunaga and Ziemann, 2010; Yeh et al., 2015), Zhang et al. (2014) reporting $C_w$ values from $0.0004 - 300$ mg m$^{-3}$ and Krechmer et al. (2016) showing values of $C_w$ to vary with $C^*$, from $C_w = 0.016$ mg m$^{-3}$ for $C^* < 1$ up to 30 mg m$^{-3}$ for $C^* > 10^4$. The reasons for the large discrepancies between studies are unknown, however are likely due to differing deformation and activities of the Teflon walls (Krechmer et al., 2016). Nonetheless comparing reported values to SOA loadings generated during the chamber experiments in this work, it is evident that gas-wall partitioning is at least competitive, if not dominant compared to gas-particle partitioning. The impact is shown in Fig. 9 by plotting the fraction of an organic species that remains in the gas-phase over different saturation vapour concentrations using $C_w = 5$ mg m$^{-3}$ and an SOA loading of 200 μg m$^{-3}$. Under this scenario gas-wall partitioning dominates, with compounds having $C^* < 10^2$ μg m$^{-3}$ predominantly residing in the walls with a small fraction in the aerosol phase after equilibrium is established, whereas species with $C^* > 10^6$ μg m$^{-3}$ remain almost entirely in the gas-phase. Compounds with $10^2 < C^* < 10^6$ μg m$^{-3}$ will partition to varying extents depending on their volatility and functional group composition between the wall, gas- and particle-phases (Krechmer et al., 2016). However no corrections for gas-particle partitioning are made in the present study, given that no product vapour loss rate measurements were made for the GIG-CAS chamber and the large variability in literature values of $C_w$. Without correcting for vapour wall losses SOA yields are likely to be underestimated (Matsunaga and Ziemann, 2010; Zhang et al., 2014; La et al., 2016)."

Figure 9 was amended to include the fraction of organic species in the gas phase for different C*. This incorporates partitioning into both SOA and the Teflon walls.

[Figure]

Figure 9. Dots show predicted first-generation and detected second-generation products from the ozonolysis of α-phellandrene in Donahue et al. (2006) space. Grey line shows the fraction of species of different saturation vapour concentrations in the gas-phase ($F_g$) after gas-wall and gas-particle equilibrium is reached, using $C_w$ = 5 mg m$^{-3}$ and an SOA loading of 200 μg m$^{-3}$. Formulation of $F_g$ is given in the Supplementary Information (S.6).

The following entry was added to the Supplementary Information.
    "S.6    Discussion of Gas-Wall Losses
The volatile species α-phellandrene, propene, acetonitrile and cyclohexane were not observed to experience losses inside the GIG-CAS reactor (this study, Wang et al., 2014). Nevertheless recent studies have shown that low volatility organic gases experience considerable losses onto reactor Teflon wall surfaces (e.g., Matsunaga and Ziemann, 2010; Zhang et al., 2014; Yeh and Ziemann, 2015; Krechmer et al., 2016; La et al., 2016). Studies have observed that organic compounds are not lost to the reactor walls, but rather partition between the gas-phase and Teflon walls in a reversible process that eventually reaches equilibrium. The time required for equilibrium depends on reactor geometry, pressure, turbulence inside the reactor and diffusion coefficients. Sorption of gaseous organic compounds to the wall and corresponding desorption from the wall back to the gas-phase can be parameterised using Raoult's law, treating the wall as a phase into which the organic compounds can partition (Matsunaga and Ziemann, 2010). The equilibrium state thus depends on compound volatility and can be modelled analogously to gas-particle absorptive partitioning, originally developed by Pankow (1994). Matsunaga and Ziemann (2010) argued that the fraction of an organic compound X that partitions into the walls at equilibrium is represented by:

$$\frac{[X]_w}{[X]_g} = K_w C_w = \frac{C_w}{C_w^*} = \frac{R\,T\,C_w}{M_w \gamma_w P_L^0} \qquad (S.6.1)$$

where $C_w$ is the equivalent organic aerosol mass concentration associated with the Teflon film, $K_w$ is the gas-wall partitioning coefficient (with

saturation concentration $C_w^*$ being the inverse) and is equal to $RT/M_w\gamma_w P_L^0$, where $M_w$ is the mean molecular mass of the Teflon film, $\gamma_w$ is the activity coefficient for the compound absorbed into the Teflon film, $P_L^0$ is the liquid vapour pressure of the compound, R is the ideal gas constant and T the temperature.

Wall loss of organics occurs in competition with gas-particle partitioning. Assuming all SOA is absorbing so that $C_{SOA}$ simply becomes [SOA] and using the parameterisation of Pankow (1994).

$$\frac{[X]_{SOA}}{[X]_g} = K_{SOA}C_w = \frac{C_{SOA}}{C_{SOA}^*} = \frac{R\,T\,[SOA]}{M_{SOA}\gamma_{SOA}P_L^0} \qquad (S.6.2)$$

The fraction of an organic compound X remaining in the gas-phase at equilibrium ($F_g$) relative to its total concentration can therefore be given by:

$$F_g = \frac{[X]_g}{[X]_{total}} = \frac{[X]_g}{[X]_g + [X]_w + [X]_{SOA}} = \frac{1}{1 + \frac{C_w}{C_w^*} + \frac{[SOA]}{C_{SOA}^*}} \qquad (S.6.3)$$

For simplicity aerosol that is lost to the walls continues to be considered as constituting the aerosol phase.

Activity coefficients for ozonolysis products of α-phellandrene in SOA generated from its decomposition were calculated using the method discussed in the main manuscript, with values ranging from 0.1 – 4. The mean molecular weight of the SOA is assumed to be 200 g mol$^{-1}$ in this work. With no available constraining information the activity coefficient of compounds absorbed into Teflon walls is assumed as 1 for simplicity (Matsunaga and Ziemann, 2010; Krechmer et al., 2016). The mean molecular mass of the Teflon film is assumed to be 250 g mol$^{-1}$ based on the masses of $-[CF_2CF_2]_n-$ and $-[CF_2CF(CF_3)]_n-$ subunits. Using these values, and given that R, T and $P_L^0$ are the same irrespective of the medium the vapour is partitioning into, the fraction of an organic compound in the gas-phase at equilibrium can be equated to:

$$F_g = \frac{1}{1 + \frac{{}^4/_5\,C_w + [SOA]}{C_{SOA}^*}} \qquad (S.6.3)$$

assuming a value of $\gamma_{SOA} = 1$. The value of $C_w$ is therefore an important parameter in determining the fraction of an organic compound that remains in the gas-phase and thus available for detection."

References Added:

Matsunaga, A. and Ziemann, P. J.: Gas-Wall Partitioning of Organic Compounds in a Teflon Film Chamber and Potential Effects on Reaction Product and Aerosol Yield Measurements, Aerosol Scie. Technol., 44, 881-892, 2010.

Yeh, G. K. and Ziemann, P. J.: Gas-Wall Partitioning of Oxygenated Organic Compounds: Measurements, Structure-Activity Relationships, and Correlation with Gas Chromatographic Retention Factor, Aerosol Sci. Technol., 49, 727-738, 2015.

Krechmer, J. E., Pagonis, D., Ziemann, P. J., and Jimenez, J. L.: Quantification of Gas-Wall Partitioning in Teflon Environmental Chambers Using Rapid Bursts of Low-Volatility Oxidized Species Generated in Situ, Environ. Sci. Technol., 50, 5757-5765, 2016.

La, Y. S., Camredon, M., Ziemann, P. J., Valorso, R., Matsunaga, A., Lannuque, V., Lee-Taylor, J., Hodzic, A., Madronich, S., and Aumont, B.: Impact of chamber wall loss of gaseous organic compounds on secondary organic aerosol formation: explicit modeling of SOA formation from alkane and alkene oxidation, Atmos. Chem. Phys., 16, 1417-1431, 2016.

Zhang, X., Cappa, C. D., Jathar, S. H., McVay, R. C., Ensberg, J. J., Kleeman, M. J., and Seinfeld, J. H.: Influence of vapor wall loss in laboratory chambers on yields of secondary organic aerosol, Proc. Natl. Acad. Scie. U.S.A., 111, 5802-5807, 2014.

*2. Page 6, 32–35: Are NO3 radicals formed under the conditions of these experiments, and if so how might that chemistry affect the results?*

Nitrate radicals are expected to form to some extent in experiment 11, where $NO_2$ is added along with $O_3$ to the reactor. However formation is only small because the reaction resulting in its formation ($NO_2 + O_3 \rightarrow NO_3$, $k = 3.2$ x $10^{-17}$ $cm^3$ molecule$^{-1}$ s$^{-1}$) competes against the much faster reaction of $O_3$ with α-phellandrene ($3.0$ x $10^{-15}$ $cm^3$ molecule$^{-1}$ s$^{-1}$) and its first-generation products ($1.0$ x $10^{-16}$ $cm^3$ molecule$^{-1}$ s$^{-1}$). If excess $O_3$ had been added to the reactor then $NO_3$ production would be expected to have a much more significant impact. Likewise if $O_3$ and $NO_2$ had been added to the reactor first and allowed to mix, the nitrate radical would have had a large impact. Reference to the nitrate radical has been added to the manuscript in response to Referee #1 (page 4 of this document).

*3. Page 7, lines 21–23: Are you certain that organic nitrates would be observed with the PTR-MS? It seems likely that these compounds could lose HNO3.*

The following change has been made to the manuscript.

p.7 l.21: "Addition of $NO_2$ to the system in experiment 11 resulted in significantly reduced yields, with overall distribution remaining similar and no new peaks or evidence of nitrate containing compounds observed. Nonetheless alkyl nitrates are known to readily lose $HNO_3$ after protonation in the PTR-TOF resulting in the formation of bare alkyl ions (D'Anna et al., 2005; Aoki et al., 2007; Duncianu et al., 2016)."

References added:

Aoki, N., Inomata, S., and Tanimoto, H.: Detection of $C_1$–$C_5$ alkyl nitrates by proton transfer reaction time-of-flight mass spectrometry, Int. J. Mass Sepctrom., 263, 12-21, 2007.

D'Anna, B., Wisthaler, A., Andreasen, Ø., Hansel, A., Hjorth, J., Jensen, N. R., Nielsen, C. J., Stenstrøm, Y., and Viidanoja, J.: Atmospheric chemistry of $C_3$-$C_6$ cycloalkanecarbaldehydes., J. Phys. Chem. A, 109, 5104-5118,

2005.

Duncianu, M., David, M., Kartigueyane, S., Cirtog, M., Doussin, J,-F., and Picquet-Varrault, B.: Measurement of alkyl and multifunctional organic nitrates by Proton Transfer Reaction Mass Spectrometry, Atmos. Meas. Tech. Discuss., in review, 2016.

*4. Page 10, lines 1–25: No mention is made of the possible effect of particle-phase reactions of O3 with first-generation products. These reactions would be very fast (re-active uptake coefficient about 0.001) and could form a variety of low volatility products.*

The parameterisation on p.10 purely investigates gas-phase chemistry. However it is true that an oversight exists in the manuscript with the possibility of heterogeneous process not discussed. This phenomenon will be explored further in the follow up publication that investigates the collected filter samples, however mention of it is now made on page 12.

p.12 l.5: "It is therefore evident that the simple mechanistic overview provided to explain formation of gas-phase products in Section 3.1.1 and in Mackenzie-Rae et al. (2016) is insufficient to account for aerosol observations, with more complex reactions or reaction processes such as autooxidation, oligomerisation and/or heterogeneous oxidation required to develop species of sufficiently low vapour pressure for both particle nucleation and growth (Hallquist et al., 2009)."

*5. Page 12, lines 26–27: Please provide a reference for the proposed relationship between density and phase state.*

Reference to the work of Kostenidou et al. (2007) has been provided.

Kostenidou, E., Pathak, R. K., and Pandis, S. N.: An algorithm for the calculation of secondary organic aerosol density combining AMS and SMPS data, Aerosol Sci. Tehnol., 41, 1002-1010, 2007.

*6. Page 12, lines 26–27: How well do the measured densities compare with those expected from measured O//C/H composition. Parameterizations have been developed for this (e.g. Kuwata et al. ES&T, 2011).*

The manuscript is updated to include a short discussion of SOA density parameterisation.

p.15 l.33: "SOA density predictions from elemental ratios using the parameterisation of Kuwata et al. (2012) show some agreement with measured values (Supplementary Information S.8)."

With the following added to the Supplementary Information.

"S.8     SOA Density Parameterisation

This study used the same method for measuring particle density and particle composition as Kuwata et al. (2012); namely comparing AMS and SMPS measurements of vacuum and mobility diameters respectively for the density and using an AMS with Aiken et al. (2008) calibration factors to measure elemental composition. The parameterisation developed by Kuwata et al. (2012) for predicting densities from elemental composition is therefore expected to be applicable. Using the Kuwata et al. (2012) parameterisation densities were predicted using elemental compositions averaged over entire experiments, with results compared to measured values in the Fig. S.8.1. Results are agreeable for most experiments except for those where the densest aerosol was produced ($\rho_{org}$ > 1.5 g cm$^{-3}$). For these, predicted density is under predicted signaling either incorrect compositional measurements or that the parameterisation is not applicable. The majority of training data for the Kuwata et al. (2012) parameterisation is for SOA with $\rho_{org}$ < 1.5 g cm$^{-3}$, with validation experiments also utilising aerosol seed particles – a notable difference compared to experiments conducted in this work. With respect to experimental measurements, the elemental ratio calibration factors of Aiken et al. (2008) do carry significant errors and recently have been superseded by Canagaratna et al. (2015). Further testing of density parameterisations is therefore recommended.

[Figure]

Figure S.8.1. Comparison of predicted to measured organic material density for α-phellandrene measured in this work (crosses, size reflect uncertainty) and α-pinene measured by Kuwata et al. (2012) (black circles). Predictions are made based on elemental composition using the parameterisation of Kuwata et al. (2012). Dashed line represents 1:1, whilst dotted lines show ±12% error representing the prediction accuracy envelope claimed by Kuwata et al. (2012). "

References added:
Kuwata, M., Zorn, S. R., and Martin, S. T.: Using elemental ratios to

predicted the density of organic material composed of carbon, hydrogen, and oxygen, Environ. Sci. Technol., 46, 787-794, 2012.

Canagaratna, M. R., Jimenez, J. L., Kroll, J. H., Chen, Q., Kessler, S. H., Massoli, P., Hildebrandt Ruiz, L., Fortner, E., Williams, L. R., Wilson, K. R., Surratt, J. D., Donahue, N. M., Jayne, J. T., and Worsnop, D. R.: Elemental ratio measurements of organic compounds using aerosol mass spectrometry: characterization, improved calibration, and implications, Atmos. Chem. Phys., 15, 253-272, 2015.

*7. Page 13, lines 5–6: The reported wall loss rates for particles seem to be much higher than those measured by others, especially for such a large chamber. Any idea why?*

The wall loss rates are higher than a lot of reported values. However many of these, especially ones reported during chamber characterisation experiments, use inert inorganic particles such as ammonium sulfate. These wall loss rates are not necessarily suitable for reactive organic aerosol (Wang et al., 2014). Wall loss rates are also dependent on turbulence inside the reactor, with two-fans operating simultaneously inside the GIG-CAS reactor during these experiments. The wall loss rates are consistent with those measured during α-pinene ozonolysis inside the chamber (Wang et al., 2014) suggesting the large loss rates are not related to specific aerosol formed but rather the GIG-CAS chamber as a whole. Meanwhile in using the same method to determine SOA wall loss rates, Pathak et al. (2007) measured wall loss rates as high as 0.48 $h^{-1}$ for α-pinene ozonolysis SOA, Pierce et al. (2008) reported SOA loss rates of up to 0.5 $h^{-1}$ for the ozonolysis of limonene and photooxidation of toluene, and Tasoglou and Pandis (2015) reported wall losses of up to 0.46 $h^{-1}$ in their study on β-caryophyllene oxidation. The values reported in this work, whilst high, are thus not thought to be unreasonable.

References:
Wang et al. Atmos. Meas. Tech., 7, 301-313, 2014.
Pathak et al. J. Geophys. Res., 112, D03201, 2007.
Pierce et al. Aerosol Sci. Technol., 42, 1001-1015, 2008.
Tasoglou and Pandis, Atmos. Chem. Phys., 15, 6035-6046, 2015.

*8. Page 13, lines 25–26: I would have expected that for a four-parameter fit would give a set a four different values. Doesn't this result imply that a two-parameter fit is as good as a four-parameter fit?*

See response to the first referee regarding the fitted parameters on page 15 of this document.

*9. Because of the large rate constant for reaction of a-phellandrene with O3 it seems likely that these experiments were conducted under conditions where RO2-RO2 reactions dominate the radical chemistry. In a clean atmosphere where this reaction might be thought to play a role in nucleation, it is likely that autoxidation may be important, or RO2-HO2 reactions. Some discussion of this would be useful.*

The following was added to the manuscript.

p.5 l.23: "A focus is on $RO_2$–$RO_2$ radical chemistry which, due to the large rate constant of α-phellandrene with ozone and lack of competing radical termination channels, dominate under the considered reaction conditions."

p.14 l.16: "Indeed the reaction conditions used in these experiments better reflects this clean environment, where reactions of $RO_2$ with $HO_2$ and other $RO_2$ radicals dominates along with unimolecular rearrangements. Such conditions favour the formation of low-volatility compounds, with the highest SOA yields for monoterpenes found under low-$NO_x$ conditions (Presto et al., 2005; Ng et al., 2007; Capouet et al., 2008; Eddingsaas et al., 2012). Under these conditions ozonolysis reactions remain important (Perraud et al. 2012; Zhao et al., 2015), which is conducive to autooxidation processes and therefore nascent SOA formation and growth due to enhanced propensity for intramolecular re-arrangements (Ehn et al., 2014; Jokinen et al., 2015). SOA yields measured in Experiment 11 however were consistent with the other ozonolysis experiments in this study (Fig. 12), suggesting that the impact of $NO_x$ on SOA yields for ozonolysis driven chemistry of α-phellandrene are limited, with sufficient condensable products still produced (Draper et al. (2015). Nonetheless the reduction in aerosol number concentration in the initial stages of experiment 11 does suggest that formation pathways of ELVOC species (i.e. oligomerisation, autooxidation) are suppressed by the inclusion of $NO_2$ (Perraud et al. 2012). Detailed modeling studies are required to establish the relative importance of α-phellandrene in different environments, although evidence suggests that it is likely a contributor to nucleation events and aerosol growth in regions where it is emitted."

References Added:
Presto, A. A., Huff Hartz, K. E., Donahue, N. M.: Secondary organic aerosol production from terpene ozonolysis. 2. Effect of NOx concentration, Environ. Sci. Technol., 39, 7046-7054, 2005.
Ng, N. L., Chhabra, P. S., Chan, A. W. H., Surratt, J. D., Kroll, J. H., Kwan, A. J., McCabe, D. C., Wennberg, P. O., Sorooshian, A., Murphy, S. M., Dalleska, N. F., Flagan, R. C., and Seinfeld, J. H.: Effect of NOx level on secondary organic aerosol (SOA) formation from the photooxidation of terpenes, Atmos. Chem. Phys., 7, 5159-5174, 2007.
Capouet, M., Müller, J.-F., Ceulemans, K., Compernolle, S., Vereecken, L., and Peeters, J.: Modeling aerosol formation in alpha-pinene photo-oxidation experiments, J. Geophys. Res., 113, D02308, 2008.
Eddingsaas, N. C., Loza, C. L., Yee, L. D., Chan, M., Schilling, K. A., Chhabra, P. S., Seinfeld, J. H., and Wennberg, P. O.: a-pinene photooxidation under controlled chemical conditions-Part 2: SOA yield and composition in low-and high-NOx environments, Atmos. Chem. Phys., 12, 7413-7427, 2012.

Zhao, D. F., Kaminski, M., Schlag, P., Fuchs, H., Acir, I. H., Bohn, B., Häseler, R., Kiendler-Scharr, A., Rohrer, F., Tillmann, R., Wang, M. J., Wegener, R., Wildt, J., Wahner, A., and Mentel, T. F.: Secondary organic aerosol formation from hydroxyl radical oxidation and ozonolysis of monoterpenes, Atmos. Chem. Phys., 15, 991-1012, 2015.

*Technical Comments*
*None.*